# Chemical and Bioactive Screening of Green Polyphenol-Rich Extracts from Chestnut By-Products: An Approach to Guide the Sustainable Production of High-Added Value Ingredients

**DOI:** 10.3390/foods12132596

**Published:** 2023-07-04

**Authors:** Daniele Bobrowski Rodrigues, Lavínia Veríssimo, Tiane Finimundy, Joana Rodrigues, Izamara Oliveira, João Gonçalves, Isabel P. Fernandes, Lillian Barros, Sandrina A. Heleno, Ricardo C. Calhelha

**Affiliations:** 1Centro de Investigação de Montanha (CIMO), Instituto Politécnico de Bragança, Campus de Santa Apolónia, 5300-253 Bragança, Portugal; daniele@ipb.pt (D.B.R.); laviniasverissimo@gmail.com (L.V.); tiane@ipb.pt (T.F.); joanapbrodrigues@ipb.pt (J.R.); izamara@ipb.pt (I.O.); lillian@ipb.pt (L.B.); calhelha@ipb.pt (R.C.C.); 2Laboratório Associado para a Sustentabilidade e Tecnologia em Regiões de Montanha (SusTEC), Instituto Politécnico de Bragança, Campus de Santa Apolónia, 5300-253 Bragança, Portugal; 3Tree Flowers Solutions, Lda, Edificio Brigantia Ecopark, Av. Cidade de Léon, 5300-358 Bragança, Portugal; joaogoncalves@treeflowerssolutions.com (J.G.); isabelfernandes@treeflowerssolutions.com (I.P.F.)

**Keywords:** *C. sativa*, biowaste, LC-MS, natural preservatives, phenolic compounds, green extraction methods

## Abstract

Opportunities for the valorisation of agro-industrial residues of the chestnut (*Castanea sativa* Mill.) production chain have been fostered with the production of multifunctional polyphenol-rich extracts with the potential to be introduced as natural additives or active components in several products. Nonetheless, it is crucial to explore the feasibility of different extracts from the various by-products for these applications through the exhaustive study of their composition and bioactivities without losing sight of the sustainable character of the process. This work aimed at the screening of the phenolic compound composition and bioactivities of different green extracts of chestnut burs, shells and leaves, as the first step to establish their potential application as natural ingredients, primarily as food preservatives. To this end, maceration (MAC) as a conventional extraction method besides ultrasound and microwave-assisted extractions (UAE and MAE) was employed to obtain the extracts from chestnut by-products using water (W) and hydroethanolic solution (HE) as solvents. Phenolic compounds were analysed by HPLC-DAD-(ESI-)MS/MS; the antioxidant capacity was assessed by colourimetric assays, and the antimicrobial activity was evaluated against several strains of food-borne bacteria and fungi. The leaf extracts obtained by MAC-HE and UAE-HE presented the highest concentration of phenolic compounds (70.92 ± 2.72 and 53.97 ± 2.41 mg.g^−1^ extract dw, respectively), whereas, for burs and shells, the highest recovery of total phenolic compounds was achieved by using UAE-HE and UAE-W (36.87 ± 1.09 and 23.03 ± 0.26 mg.g^−1^ extract dw, respectively). Bis-HHDP-glucose isomers, chestanin and gallic acid were among the most abundant compounds. Bur extracts (MAC-HE and UAE-HE) generally presented the highest antioxidant capacity as measured by TBARS, while the best results in DPPH and reducing power assays were found for shell extracts (MAE-W and MAC-HE). Promising antibacterial activity was noticed for the aqueous extracts of burs, leaves and hydroethanolic extracts of shells, with emphasis on the MAE-W extract of burs that showed bactericidal activity against *E. cloacae*, *P. aeruginosa* and *S. aureus* (MBC 5 mg.mL^−1^). Overall, it can be concluded that chestnut by-products, including burs, shells and leaves, are sources of polyphenolic compounds with significant antioxidant and antimicrobial activities. The choice of extraction method and solvent greatly influenced the composition and bioactivity of the extracts. These findings highlight the potential of chestnut by-products for the development of natural additives, particularly for food preservation, while also emphasizing the importance of sustainable utilization of agricultural waste materials. Further research is warranted to optimize extraction techniques and explore additional applications for these valuable bioactive compounds.

## 1. Introduction

Global efforts are being directed towards promoting renewable feedstocks as primary sources of chemical compounds. Scientific entities and industries are being increasingly called upon to consider naturally derived chemicals, particularly those obtained through the circular economy and biorefinery platforms, as a viable means for creating a more sustainable and healthier society [1]. This aligns with the United Nations’ Sustainable Development Goals, closely linked to the principles of Green Chemistry and Food Bioeconomy [2]. Indeed, natural extracts derived from residual agrifood by-products through environmentally friendly technologies are envisioned as an important source of biomolecules of interest for diverse industrial applications.

In the context of Food and Beverage companies, the demand for natural ingredients has become a decisive factor supporting consumers’ choices [3,4]. This has been driven, at least in part, by the growing understanding of the link between the quality of food intake and health. Scientific evidence is pointing out that synthetic additives, such as preservatives used to extend the shelf-life of products, as being associated with allergies and other potential health risks [5]. Additionally, consumers’ perception that dietary choices may critically impact the environment has led to an accelerated search for clean-label food products, containing natural and functional ingredients obtained from sustainable backgrounds [6]. As a result, investments are directed at finding, producing and launching natural and sustainable alternatives to synthetic ingredients that can meet consumer demand while ensuring product safety and quality.

Natural extracts from agro-industrial residues of the chestnut production chain have grounded attention as promising alternative ingredients with functional and technological properties for several industrial sectors [7,8]. Native from the northern hemisphere’s temperate zones, the chestnut (*Castanea sativa* Mill.), also referred to as the sweet or European chestnut, is one of the most traditional and economically important local commodities [9]. Chestnut trees have been typically cultivated since ancient times because of their wood and edible nuts, a sensory-appreciated food source of high nutritional value used to prepare several local dishes and desserts [10]. In addition to direct consumption, a growing proportion of crop production is used in the food industry to produce chestnut-based products such as flour and purée or as a gluten-free ingredient [11,12]. At the same time, however, this production chain generates massive quantities of by-products and residues that can hold economic, environmental, and management costs [13,14]. These include the chestnut burs, a dome of spiny bracts containing the nuts, flowers, and leaves, besides the inner and outer parts of the chestnut shells (integument and pericarp), removed during industrial processing [15]. In this regard, recovering biomolecules from plentiful and locally available chestnut plants’ residual parts can be a valuable strategy to reduce the environmental impact of agrifood sector activities and aggregate value to by-products, effectively structuring the production chain [7,14].

Recent in vitro and in vivo studies have shown the potential application of chestnut by-products as active components for pharmaceutical, nutraceutical, and cosmetical formulations [16,17,18,19,20,21]. Works from our research group have focused on food applications of male chestnut flowers and their extracts, and the obtained results pointed out an efficient preservative role when applied to artisanal or processed products to which they are added, including traditional Portuguese cheese and pastry goods ([22,23,24]. Herein, Caleja et al. (2020) [25] demonstrated the viability of applying chestnut flower aqueous extract as a natural preservative in Pasteis de nata to replace potassium sorbate. Of particular relevance, Heleno et al. (2020) [26] developed and patented a chestnut flower-based preservative named ChestWine^®^ that has been incorporated in commercially available wines as synthetic sulphites replacer (https://casadojoa.com/vinhos/feito-de-joa/ (accessed on 17 January 2023)). The potential use of this by-product as a functional agent has also been explored [27]. From that perspective, questions have been raised on the feasibility and performance of other chestnut by-products for these industrial applications.

The beneficial properties derived from chestnut by-products, in terms of either human and animal health or food systems, have been attributed to the phytochemical profile of this botanical matrix, mainly due to the combination of phenolic compounds present in *C. sativa* [28,29,30,31,32,33]. These plant metabolites are widely accepted as bioactive compounds and are consistently associated with antioxidant and antimicrobial activities. By inhibiting or attenuating microbial proliferation and oxidative processes deleterious to food biomolecules, natural polyphenols are hypothesized to enhance product stability, quality, and shelf-life [34,35,36]. Nevertheless, obtaining natural-based extracts and ingredients to meet the crescent demand of the market depends on the efficient extraction of these compounds. Conventional extraction methods are generally time-consuming and often involve the use of organic solvents not allowed for food applications. Therefore, more efficient extraction systems are required to recover natural biomolecules from their matrices, such as those based on emerging technologies like microwaves and ultrasounds, recognized for bearing a series of compromises in regard to environmental and economic aspects [2,37]. Notably, the extraction procedure should consider not only the extraction yields but also the bioactive efficiency of the obtained extracts and the process sustainability [38].

Even though the characterization of chestnut by-products has already been investigated, a significant limitation is the general lack of quantitative information provided by selective and sensitive analytical methodologies of choice, such as hyphenated techniques involving chromatographic separation and UV-Visible (UV-Vis) or mass spectrometry (MS) detection. Few studies have directly compared extracts from different chestnut by-products, with most of them relying either on a single or conventional extraction condition, estimates using non-specific spectrophotometric assays, or solely on the quantitation of particular compounds in the extract [39,40,41,42]. From this perspective, this work aimed at systematically screening the phenolic composition and bioactivities of extracts of *C. sativa* leaves, burs and shells produced using different extraction conditions. For this purpose, food-grade GRAS (*Generally Recognized as Safe*) solvents were selected, as well as two of the most mature green extraction techniques in terms of academic applications and a conventional method.

## 2. Material and Methods

### 2.1. Chemicals

Standards of phenolic compounds: Ellagic acid, Gallic acid, Quercetin-3-*O*-glucoside and Epicatechin (purity ≥ 90%, HPLC) were purchased from Extrasynthése (Genay, France) and Sigma-Aldrich (St. Louis, MO, USA). Analytical grade reagents trichloroacetic acid (TCA), Trolox, Tris, ascorbic acid, ellipticine and dimethyl sulfoxide (DMSO) were purchased from Sigma-Aldrich (St. Louis, MO, USA). Thiobarbituric acid (TBA), Tryptone Soy Broth (TSB), *p*-iodonitrotetrazolium (INT), chloride sodium sulphate, calcium chloride and magnesium chloride were purchased from Panreac Applichem (Barcelona, Spain), whereas iron (II) sulphate was acquired from ACROS Organics (Geel, Belgium). Potassium dichromate, sodium bicarbonate and magnesium sulphate were supplied by Merck (Darmstadt, Germany), and potassium chloride was purchased from Pronalab (Lisbon, Portugal). The reagent 2,2-diphenyl-1-picrylhydrazyl (DPPH) was obtained from Alfa Aesar (Ward Hill, MA, USA). Porcine (Sus scrofa) brain used in antioxidant activity was obtained from official slaughtering animals. Microbiology supplies such as Malt extract broth (MEB), blood (sheep blood 7%), and MacConkey agars were purchased by LiofilChem S.R.L (Roseto d. Abruzzi, TE, Italy). Antibiotics methicillin, streptomycin and ampicillin used for control assays were supplied from Fisher Scientific (Janssen Pharmaceutical, Belgium), whereas the antifungal ketoconazole was provided from Frilabo (Porto, Portugal). For cell culture, Dulbecco’s modified Eagle’s (DMEM) and Roswell Park Memorial Institute (RPMI 1640) media, Hank’s balanced salt solution (HBSS), fetal bovine serum (FBS), trypsin–EDTA, *L*-glutamine, penicillin and streptomycin were acquired from Hyclone (Logan, UT, USA). HPLC-grade methanol, ethanol, acetonitrile and extra pure formic acid were provided by Fisher Scientific (Leicestershire, UK). Water was purified in a TGI Pure Water Systems (Greenville, SC, USA). HPLC solvents and samples were filtered through 0.45 and 0.22 μm membranes, respectively, prior to chromatographic analysis.

### 2.2. Plant Material

Samples of by-products of *C. sativa* Mill. (leaves, burs and shells) were collected in the year 2022 from a chestnut orchard of the Longal variety, located in the Northeast of Portugal, at Vinhais municipality (coordinates 41°50′11.98″ N, 7°9′23.695″ W). Batches of at least two kilos of each sample were collected already dried in the field and transported to the laboratory. After, they were reduced to a fine powder to produce a homogeneous sample of each by-product, which was stored at room temperature, protected from light, until extraction procedures.

### 2.3. Extraction of Phenolic Compounds

#### 2.3.1. Maceration (MAC)

For the traditional solid–liquid extraction technique, aliquots of 2 g of each dried sample were weighed and mixed with 60 mL of 80% ethanol (ethanol:water, 80:20, *v*/*v*) at room temperature. The mixture was continuously stirred (500 rpm, 1 h at room temperature), using a magnetic bar in a stirrer plate (Multimatic 9-N, Selecta—Barcelona, Spain). Upon completion of this process, the extract was filtered (filter paper, Ø 125 mm, CMHLAB—Barcelona, Spain). The sample residue was reextracted under the same conditions, and both extracts were combined. Extracts of chestnut by-products obtained by maceration with hydroethanolic solvent are referred to as MAC-HE throughout the text.

#### 2.3.2. Ultrasound-Assisted Extraction (UAE)

UAE, a promising alternative to traditional extraction techniques, was carried out using an ultrasonic device (model CY-500, Optic Ivymen System—Barcelona, Spain) operating at 25 kHz and equipped with a titanium probe. A portion of 3 g of each sample was added to 100 mL of either 80% ethanol or ultrapure water and subjected to sonication for 27.3 ± 2.6 min at a power of 235.8 ± 36.8 W using an ice bath to avoid overheating of extractive mixtures [43]. UAE extracts of chestnut by-products prepared using either hydroethanolic solution or water as extraction solvents are henceforth called UAE-HE and UAE-W, respectively.

#### 2.3.3. Microwave-Assisted Extraction (MAE)

Also considered an innovative extraction approach, MAE was conducted using an analytical, enclosed microwave system (Speedwave Xpert, Berghof—Eningen, Germany). The process involved the extraction of 3 g of each sample, separately, in 100 mL of ultrapure water, in a closed vessel for 2 min at 80 °C (microwave power, P, 50–1000 W). The extracts obtained from different parts of the *C. sativa* plant are referred to as MAE-W in this study.

After each extraction procedure described above, aqueous and hydroethanolic extracts rich in phenolic compounds were separated from the solid residues by filtration (filter paper, Ø 125 mm, CMHLAB—Barcelona, Spain). Aqueous extracts (UAE-W and MAE-W) were frozen and subjected to freeze-drying (Freeze Dryer Telstar LyoQuest-55—Milan, Italy) for 48 h at −55 ± 0.5 °C, while the hydroethanolic extracts (MAC-HE and UAE-HE) ethanol was first evaporated under vacuum (T < 37 °C, rotatory evaporator Büchi R-210—Flawil, Switzerland) and then the remaining aqueous portion was likewise subjected to freezing and freeze-dried under the same aforementioned conditions.

### 2.4. Composition in Phenolic Compounds

Extracts were analysed in a Thermo Scientific high-performance liquid chromatograph (HPLC—Dionex UltiMate^™^ 3000 series, Thermo Fisher Scientific—San Jose, CA, USA) equipped with a diode array detector (DAD) and connected in series to an Orbitrap mass spectrometer (MS, Orbitrap Exploris^™^ 120, Thermo Fisher Scientific—San Jose, CA, USA). Phenolic compounds were separated in a Spherisorb S3 ODS-2 C_18_ column (3 μm, 4.6 mm × 150 mm, Waters—Milford, CT, USA) kept at 35 °C, under a gradient of 0.1% (*v*/*v*) formic acid in ultrapure water (A) and acetonitrile (B). The mobile phase proportion (A:B, %) ranged from initial 85:15 for 5 min to 80:20 in 5 min, reaching 75:25 in 10 min, 65:35 in 10 min and 50:50 in 10 min, finally returning in 10 min to the initial condition, kept for further 10 min for column reconditioning. The flow rate was 0.5 mL.min^−1^, and the Injection volume 10 μL. UV-Vis spectra were acquired between 180 to 700 nm, and the chromatograms processed at 280, 330 and 370 nm for the different classes of phenolic compounds. The HPLC eluate was analysed by high-resolution, tandem mass spectrometry, and the compounds were ionized using an OptaMax NG electrospray ion source (ESI) source operating in negative mode. The spray voltage was set at 2.5 kV, the ion transfer tube temperature at 325 °C, and the vaporizer temperature at 350 °C. Nitrogen served as the sheath gas (50 arb), auxiliary gas (10 arb) and sweep gas (1 arb). Full MS and MS/MS spectra were acquired in the range from 110 to 1800 charge-to-mass ratio (*m*/*z*), with a resolution of 15,000 and RF lens kept at 70%. The full scan MS was followed by the top four data-dependent MS^2^ (ddMS^2^) scans acquired by applying an HCD (high-energy collisional dissociation) normalized at 30% in the stepped (30, 50 and 150) mode. When needed, a dynamic exclusion strategy was employed. Data acquisition and processing were conducted with the Xcalibur^™^ software (Thermo Fisher Scientific, San Jose, CA, USA). For compound identification, elution order on the C_18_ column and characteristics of the UV-Vis and mass spectra (molecular ion ([M-H]^−^), and MS/MS fragments) were interpreted and compared with standards when available, literature data and libraries (NIST^™^, MZ Vault^™^ and MZCloud^™^) available in the Freestyle^™^ 1.7 software (Thermo Fisher Scientific, San Jose, CA, USA), also employed for data processing. Quantification was performed using 9-point external calibration curves of authentic standards of gallic acid (1–89 μg.mL^−1^), ellagic acid (1–45 μg.mL^−1^), epicatechin (1–85 μg.mL^−1^) and quercetin-3-*O*-glucoside (0.1–59 μg.mL^−1^). Phenolic compounds identified for which a standard was unavailable were estimated through the analytical curve of the most similar standard available. The results of phenolic compounds were expressed as mg per g of freeze-dried extract (mg.g^−1^, dry weight, dw).

### 2.5. In Vitro Bioactivity Assays

#### 2.5.1. Antioxidant Capacity

The antioxidant capacity of the different extracts of chestnut by-products was assessed by three distinct in vitro methodologies, namely Thiobarbituric Acid Reactive Species (TBARS), 2,2-diphenyl-1-picrylhydrazyl (DPPH^•^) radical scavenging and reducing power (RP) assays.

##### TBARS Assay

The capacity of the extracts of chestnut by-products to inhibit lipid peroxidation as estimated by the inhibition of the formation of thiobarbituric acid reactive substances was assessed using porcine brain homogenates [44]. For that, a given mass of the pig brain was weighed and twice as much (*w*/*w*) of Tris-HCl buffer (20 mM, pH 7.4) was added to the reaction tube. The mixture was homogenized and centrifuged (3500 rpm, 10 min) to obtain a liquid tissue homogenate as supernatant. Freeze-dried extracts were suspended in their extraction solvents and a serial dilution of the extract solution was carried out, starting from the concentration of 20 mg.mL^−1^. A total of 12 distinct concentrations were prepared to enable finding the linear range of activity of each extract and, consequently, which concentration inhibits 50% of the reaction. A volume of 200 µL of each extract concentration, in triplicate, was pipetted into a 48-well microplate, followed by the addition of 100 µL of 0.1 mM ascorbic acid, 100 µL of 10 mM iron sulphate and 100 µL of the pig brain homogenate. The microplate was incubated (37 ± 0.5 °C, 1 h), and in the sequence, 500 µL of 28% (*w*/*v*) trichloroacetic acid and 380 µL of 2% (*w*/*v*) thiobarbituric acid, both freshly prepared, were added. The plate was further incubated (80 ± 0.5 °C, 20 min) and the contents of each well were transferred to 2 mL tubes, which were centrifuged (3000 rpm, 5 min). Supernatants of each tube were transferred to wells of a 96-well plate, and the absorbance was measured at 532 nm (SPECTROstar Nano spectrophotometer, BMG LABTECH—Ortenberg, Germany). Trolox, a hydrophilic analogous to vitamin E, was used as the positive control, and the extraction solvent of each extract was used in the blank assay. The percentage of inhibition of the lipid peroxidation was calculated as follows:(1)$$I\left(\%\right)=\frac{Abs_{control}-Abs_{ext}}{Abs_{control}}\times 100,$$
where the absorbance presented by the blank assay (negative control) is referred to as *Abs_control_* and that presented by a given extract concentration is referred to as *Abs_ext_*. By analysing the relation between the concentrations tested and their respective percentage of inhibition, the results were expressed as EC_50_ values, i.e., the effective concentration that provides a half-maximal antioxidant response (μg.mL^−1^), or in other words, the concentration of extract able to inhibit lipid peroxidation by 50%.

##### DPPH^•^ Radical Scavenging Assay

The DPPH assay was conducted according to the classical method of Brand-Williams et al. (1995) [45]. A 20 g.mL^−1^ solution of each by-product extract was prepared by dissolving the freeze-dried powder in their extraction solvent and subjected to a serial dilution to prepare 8 different extract concentrations, which were pipetted, in triplicate, to 96-well microplates. A methanolic 6 × 10^−5^ M solution of DPPH^•^ radical was prepared, and a volume of 270 µL was transferred to wells already containing 30 µL of either each by-product extract, extraction solvents (blank assays) or Trolox solution (positive control). After 1 h incubation (room temperature, absence of light), the microplate was taken for absorbance reading in SPECTROstar Nano spectrophotometer (BMG LABTECH, Ortenberg, Germany) at 515 nm. The results were also expressed in terms of EC_50_ (μg.mL^−1^) and determined similarly to that applied for the TBARS assay.

##### Reducing Power Assay

The same extract serial dilutions used in the DPPH method were subjected to the RP assay. Aliquots of 0.5 mL of those different extract concentrations were mixed with the same volume of both sodium phosphate buffer (200 mM, pH 6.6) and potassium ferricyanide (1%, *w*/*v*). The reaction tubes were incubated at 50 °C for 20 min, and then 0.5 mL of 10% trichloroacetic acid was added to the mixtures. A fraction of them (0.8 mL) was transferred to 48-well plates and added the same amount of ultrapure water and 0.1% ferric chloride (160 μL). The absorbance of those solutions was recorded at 690 nm using a SPECTROstar Nano spectrophotometer microplate reader (BMG LABTECH, Ortenberg, Germany). Trolox was used as a standard. The extract concentration providing 50% of the maximum antioxidant activity (EC_50_, μg.mL^−1^) was then calculated according to Equation (2).
(2)$$I\left(\%\right)=\frac{Abs_{ext}-Abs_{control}}{Abs_{ext}}\times 100,$$
with *Abs_control_* and *Abs_ext_* referring to the absorbance measured in the blank assay and a certain extract concentration, respectively.

#### 2.5.2. Hepatotoxicity

The hepatotoxic activity of the phenolic compound extracts was assessed against a primary cell line PLP2 (hepatocytes from pig liver slaughtered in a local abattoir) according to the methodology of Mandim et al. (2018). Primary cell maintenance was performed in RPMI-1640 medium enriched with 10% foetal bovine serum, 2 mM of glutamine and antibiotics penicillin (100 U.mL^−1^) and streptomycin (100 mg.mL^−1^). The culture flasks were incubated at 37 ± 0.5 °C in a humid atmosphere with 5% CO_2_, and the experiments were carried out using cultures presenting 70 to 80% confluence. A stock solution (8 mg.mL^−1^, *w*/*v*) of each freeze-dried extract in sterile, ultrapure water was prepared, from which serial dilutions were made (ranging from 8.0 to 0.125 mg.mL^−1^). In 96-well microplates, 10 μL of each extract concentration and 190 μL of the cell line suspension (corresponding to 1.0 × 10^4^ cells/well) were added in duplicates. After the cell adherence was ensured, the microplates were incubated (37 ± 0.5 °C for 72h in a humid environment with 5% CO_2_) and upon completion of incubation 100 μL of 10% (*w*/*v*) cold trichloroacetic acid was added to stop the reaction. The microplates were then further incubated (4 ± 0.5 °C for 1 h), washed under tap water and, after drying, 100 μL of the reagent SRB (0.057%, *w*/*v*) was added and the plates were kept for 30 min at room temperature. The excess of SRB that has not adhered was washed out with 1% (*v*/*v*) acetic acid. In contrast, the adhered reagent was solubilised with 200 μL of 10mM Tris and estimated through absorbance reading at 540 nm (ELX800 microplate reader—Bio-Tek Instruments, Winooski, VT, USA). Ellipticine was used as the positive control. The results were expressed as growth inhibition 50% (GI_50_), referring to the extract concentration able to inhibit the maximum cell growth by half, which was estimated similarly to those results of the TBARS assay.

#### 2.5.3. Antimicrobial Activity

The antibacterial and antifungal activities of the extracts were determined according to methods described in detail by Pires et al. (2018) [35] and Heleno et al. (2013) [46], respectively. Five Gram-negative bacteria (*Enterobacter cloacae*—ATCC 49741, *Escherichia coli*—ATCC 25922, *Pseudomonas aeruginosa*—ATCC 9027, *Salmonella enterica*—ATCC 13076, and *Yersinia enterocolitica*—ATCC 8610) and three Gram-positive bacteria (*Bacillus cereus*—ATCC 11778, Listeria monocytogenes—ATCC 19111 and *Staphylococcus aureus*—ATCC 25923), besides two fungi strains (*Aspergillus fumigatus*—ATCC 204305 and *Aspergillus brasiliensis*—ATCC 16404) purchased from Frilabo (Porto, Portugal) were used. All work was performed with sterile materials handled under laminar flow. Prior to the assays, microorganisms were incubated (37 °C ± 0.5 °C for 24 h for bacteria and 25 °C ± 0.5 °C for 72 h for fungi) with appropriate media for each strain to reach a state of exponential growth. Negative controls of the extract and culture medium were prepared, and the antibiotics streptomycin, methicillin, ampicillin and antifungal ketoconazole served as positive controls. The antimicrobial potential was assessed as the minimum concentration required to inhibit the microorganism growth (MIB) and to cause bacteria or fungi death (MBC or MFC, respectively).

### 2.6. Statistical Analyses

For screening and ranking purposes, the means of total phenolic contents of all the twelve extracts were thoroughly compared by analysis of variance (ANOVA) with 2 variables (chestnut by-product and extraction procedure), followed by Tukey’s post hoc test (α = 5%). The analysis of the quantitative data of phenolic compounds showed a significant interaction between the two factors (*p* < 0.05). This analysis was carried out using the Statistica 7.0 software, while the regression analyses, Principal Component Analysis (PCA), and the construction of chromatograms were carried out using Origin 8.5 Software.

## 3. Results and Discussion

### 3.1. Composition of Polyphenols in Extracts of Chestnut By-Products

#### 3.1.1. Phenolic Compound Identification

The analysis of phenolic compounds in the extracts of *C. sativa* by-products by HPLC-DAD-(ESI)MS/MS showed the presence of a total of 39 different compounds, including regioisomers, being 12 identified in shells, 19 in burs, and 22 in leaves (Table 1, Figure 1). Interestingly, only one compound among those identified (gallic acid, peak 1) was common to all three by-products. Apart from that phenolic acid, shells shared only a single isomer of galloyl-bis-hexahydroxydiphenoyl(HHDP)-glucose (peak 3) with leaves and none compound with burs. Bur and leaf extracts, on their turn, presented several compounds in common, being 14 out of the 19 phenolic compounds found in burs also present in leaves. In other words, shells displayed the most different chemical profile among the three chestnut parts evaluated (Figure 1). This is consistent with the data provided by Silva et al. (2020) [41], which compared the polyphenol profile of conventional ethanolic extracts from *C. sativa* by-products. Their results showed a higher number of compounds in chestnut leaf extracts, the majority of which was also identified in burs, while chestnut shells presented the most pronounced difference in phenolic profile. On the other hand, they found a total of 14 different compounds, which is half the number of compounds found in the present investigation.

Among the polyphenols found in the extracts, seven were identified as phenolic acids and their *O*-substituted derivatives. Gallic acid (peak 1) and ellagic acid (peak 19), both benzoic acid derivatives, were positively identified as they displayed retention time and spectroscopic characteristics identical to the reference standards available. With characteristic early elution on the reverse phase column, gallic acid presented a UV-Vis spectrum with two ranges of maximum absorbance, around 210 and 270 nm, besides the molecular ion [M-H]^−^ at *m*/*z* 169, which generated abundant fragment ion [M-H-CO_2_]^−^ at *m*/*z* 125 in the MS^2^ spectrum due to decarboxylation (Figure 2). In addition, ellagic acid absorbed maximally around 365 nm and presented the ion at *m*/*z* 301 ([M-H]^−^) in the MS spectrum, besides typical MS^2^ fragments at *m*/*z* 257 and 229 (Bowers et al., 2018) [47]. This phenolic acid occurs in nature mainly in its esterified form, either as ellagitannins or simple glycosides, being nuts and their by-products one of the primary sources of ellagic acid in the diet [48]. Indeed, methylated, and glycosylated derivatives of ellagic acid were also found in chestnut by-products, as reported by other studies [49,50]. Compounds from peaks 34, 37 and 38 were identified as mono-methyl ellagic acid glycosides since presented a similar pattern of light absorption and generated the same MS^2^ ions at *m*/*z* 315 due to the loss of a sugar moiety and at *m*/*z* 301, corresponding to the further loss of a methyl group (−15 u) and to the ellagic acid molecule. For peak 34, assigned as methyl ellagic acid hexoside, the ion at *m*/*z* 315 corresponded to the neutral loss of a hexosyl residue (−162 u) after the fragmentation of the molecular ion [M-H]^−^ at *m*/*z* 477. Likewise, the peaks 37 and 38 that shared the molecular ion [M-H]^−^ at *m*/*z* 461 showed ions [M-H-deoxyhexosyl]^−^ at *m*/*z* 315 and [M-H-deoxyhexosyl-CH_3_]^−^ at *m*/*z* 301 after fragmentation, allowing their tentative identification as methyl ellagic acid deoxyhexoside isomers. On the other hand, peaks 35 and 39 displayed the ion at *m*/*z* 343 as their base peak in MS/MS spectra, consistent with a three-methylated ellagic acid structure after losing a hexosyl moiety (−162 u) from the molecular ion [M-H]^−^ at *m*/*z* 505. Using the dynamic exclusion tool, the additional cleavage of the main fragment yielded ions related to sequential losses of methyl groups (15 u) and the backbone of ellagic acid. Therefore, those peaks were tentatively identified as trimethyl-ellagic acid hexoside isomers. The most prominent ion detected in the MS spectra for those isomers was the formic acid adduct [M-H+HCOOH]^−^ at *m*/*z* 551, in line with the findings of Formato et al. (2022) [50] that has isolated this compound and analysed it by NMR.

In addition to gallic and ellagic acids, gallotannins and ellagitannins were also detected in chestnut by-products. These molecules comprise the so-called group of hydrolysable tannins as they are readily hydrolysed to release gallic and/or HHDP acids, the latter being spontaneously converted into ellagic acid. Hydrolysable tannins are esters of these acids with a polyol, primarily glucose. As the substitutes may be attached to different hydroxyl groups in the polyol core and tend to form oligomers, several regioisomers and compounds that are structurally related can be found in nature, as seen in Table 1.

Isomers of five gallotannin representatives were found in extracts from leaves and burs, precisely molecules composed of hexose, 3,4,5-trihydroxy benzyl alcohol and gallic or dehydrodigallic acid units (Figure 3). First isolated and elucidated by Osawa and collaborators in chestnut, this group of gallotannins was distinguished by fragment ions corresponding to gallic acid (*m*/*z* 169) and to the neutral loss of the terminal trihydroxy benzyl alcohol-hexoside residue (−300 u, dehydrated) [50,51,52]. This latter structure was referred to as crenatin (MW 317) by the same authors, one of the major compounds identified in Italian PGI chestnut “Marrone di Roccadaspide”, although not detected in the present study [51,53,54]. Nonetheless, its analogous galloylated structure, cretanin (peak 16) was identified based on its molecular ion [M-H]^−^ at *m*/*z* 469, which yielded an intense fragment ion [M-H-300]^−^ at *m*/*z* 169, consistent with a gallic acid molecule upon cleavage and released of the hexosyl-trihydroxy-benzyl alcohol unit (Formato et al., 2022; Cerulli et al., 2021). In accordance with previous studies, this compound was not detected in shells [8,53]. Additionally, chesnatin isomers (peaks 4 and 11) were detected in leaves and displayed a molecular ion [M-H]^−^ at *m*/*z* 637 and an abundant fragment ion at *m*/*z* 467 corresponding to the galloyl-trihydroxybenzyl-hexoside structure, as the result of the elimination of a gallic acid group (−170 u), besides the ions at *m*/*z* 305 and *m*/*z* 169 produced by the respective and consecutive losses of the hexosyl and the trihydroxybenzyl alcohol residues from the remaining gallic acid backbone [55]. With a molecular ion [M-H]^−^ at *m*/*z* 937, peaks 17 and 26 detected in burs and leaves were assigned as chestanin isomers. Indeed, the daughter ions [M-H-300]^−^ at *m*/*z* 637, [M-H-300-170]^−^ at *m*/*z* 467 (base peak), [M-H-300-170-162]^−^ at *m*/*z* 305 and [M-H-300-170-300]^−^ at *m*/*z* 169 indicated the occurrence of one more hexosyl moiety and one more trihydroxybenzyl alcohol unit than chesnatin [56] (Figure 3). All these compounds, as simple glucosides or esters of gallic acid, presented UV-Vis spectra that resemble that of this acid.

Distributed by all the chestnut by-products, eleven ellagitannins were tentatively identified based on their characteristic fragmentation pattern, with neutral losses of one or more units of HHDP (−302 u) and, in some cases, gallic acid (−170 u or −152 u for dehydrated galloyl moieties), often accompanied by the loss of sugar (usually glucose, loss of −180 u or −162 u for the dehydrated residue) [57]. For instance, the elimination of structures galloyl-glucose (−332 u), HHDP-glucose (−482 u) and galloyl-HHDP-glucose (−634 u), typical of this group of compounds were noticed, yielding diagnostic ion at *m*/*z* 301 suggestive of the presence of HHDP in the molecule. Fragments at *m*/*z* 245 and 279, recently described as reporter ions for the screening of ellagitannins containing HHDP, were also detected in the MS^2^ spectra [52] (Figure 2). The shape of the UV-Vis spectrum of these compounds provided information on the proportion of free and bound galloyl units in their structure and was also considered for the peak assignment. Typically, in the presence of free galloyl moieties, the spectrum is similar to that of gallic acid, with a visible valley between the two spectral regions of maximum absorbance (Figure 2). As the quantity of interlinked galloyl or HHDP groups in the molecule increases, the valley gradually disappears until the spectrum shape becomes a simple slope when there is no free galloyl [57].

Peaks 2 and 9 shared the same molecular ion [M-H]^−^ at *m*/*z* 783 and MS^2^ ions [M-H-302]^−^ at *m*/*z* 481, evidencing the loss of an HHDP, and [M-H-302-180]^−^ at *m*/*z* 301 corresponding to the loss of an HHDP-glucose group (482 u) and to the ellagic acid. The complete absence of a valley between the two bands of maximum absorbance in their UV-Vis spectra was consistent with the absence of free galloyl units [57] (Figure 2). These characteristics, along with their early elution on the reverse phase, allowed their identification as bis-HHDP-glucose isomers, also called pedunculagin I. Peaks 3, 7, 14, 15 and 22 presented the molecular ion [M-H]^−^ at *m*/*z* 935 and fragment ions [M-H-302]^−^ at *m*/*z* 633 and [M-H-302-332]^−^ at *m*/*z* 301 in their MS/MS spectra. These neutral losses indicated the occurrence of one more galloyl unit than pedunculagin eliminated as a galloyl-HHDP-glucose unit (−634 u). Their UV-Vis spectra showed a slightly visible valley, also suggesting a reduced HHDP:galloyl ratio in the molecule in contrast to the same compound (Figure 2). Therefore, these peaks were tentatively identified as galloyl-bis-HHDP-glucose isomers. With a molecular ion [M-H]^−^ at *m*/*z* 785 and MS^2^ fragment ions at *m*/*z* 633, 483 and 301 due to the successive losses of two galloyl moieties (−152 u) and one glucose (−180 u), peaks 5 and 8 were tentatively identified as digalloyl-HHDP-glucose isomers. The molecular ion [M-H]^−^ at *m*/*z* 937 and MS^2^ fragment ions at *m*/*z* 637 and 301 found in peak 6 were consistent with those of compounds previously identified in chestnut catkins as trigalloyl-HHDP-glucose [23,24,29]. Their pattern of light absorption was coherent with the presence of three galloyl to one HHDP group. Finally, peak 23 presented molecular ion [M-H]^−^ at *m*/*z* 933, yielding fragment ions at *m*/*z* 631 (−302 u), 601 (gallagyl moiety), 451 (−180 u) and 301 (−150 u). In fact, differently from the previous ellagitannins, this compound displayed a UV-Vis with maximum wavelengths of absorption resembling that of the ellagic acid and consistent with the presence of the gallagyl chromophore. Based on these features, the compound was tentatively identified as galloyl-gallagyl-hexoside also called galloyl-punicalin [58,59]. Whereas the previous ellagitannins contained HHDP acid or their derivatives and formed ellagic acid upon their release and conversion, this latter structure contains ellagic acid.

Unlike the hydrolysable tannins distributed through all the extracts analysed, the class of condensed tannins was exclusively detected in shells. Furthermore, the molecular ions indicated relatively high molecular weights, these compounds were assigned as procyanidin oligomers based on a series of ions in their MS^2^ spectra differing by multiples of 288 u. This structural feature is consistent with cleavages between monomeric subunits of flavan-3-ols ((*epi*)catechin units) linked by interflavonoid C-C bonds. In addition, they presented strong absorption in the UV region of the electromagnetic spectrum (~280 nm) as observed for condensed tannins [60,61]. Peak 10, identified as a procyanidin tetramer, presented a molecular ion [M-H]^−^ at *m*/*z* 1153 and fragments at *m*/*z* 865, 577, 575 and 289, characteristic quinone methide (QM) fragmentation pattern of a B-type tetrameric procyanidin [61,62]. Peaks 13 and 32 presented the respective molecular ions [M-H]^−^ at *m*/*z* 865 and 867 and were assigned as type-B procyanidin trimers.

Flavonoids represented the second family of phenolic compounds in the number of compounds among the samples (Table 1). As for the hydrolysable tannins, the building block of condensed tannins, the flavan-3-ol(-)-(*epi*)catechin (peak 12), was also detected in the present study. This compound was found exclusively in shells and identified by comparison with the reference standard, with which shared retention time, UV-Vis and mass spectral characteristics. The remaining flavonoids corresponded primarily to flavonol derivatives as also described elsewhere [23,29], with quercetin-*O*-glycosylated derivatives standing out with seven compounds identified. Quercetin-3-*O*-rutinoside (peak 20) and quercetin-3-*O*-glucoside (peak 25) also had their identity confirmed by comparison of their elution, mass and UV-Vis characteristics compared with reference standards. All the quercetin glucosides absorbed light maximally around 355 nm and presented abundant ion at *m*/*z* 301 corresponding to the aglycone in the MS^2^ spectra upon neutral losses of sugar residues from the molecular ion. In this regard, peaks 21 ([M−H]^−^ at *m*/*z* 609), 23 ([M−H]^−^ at *m*/*z* 477), 27 ([M−H]^−^ at *m*/*z* 593), 28 ([M−H]^−^ at *m*/*z* 433) and 31 ([M−H]^−^ at *m*/*z* 447) presented a neutral loss of a deoxyhexosyl-hexoside (308 u), a glucuronyl unit (176 u), two rhamnosyl moieties (146 u), a pentose (132 u), and deoxyhexoside (146 u), being tentatively identified as quercetin-deoxyhexosyl-hexoside, quercetin-3-*O*-glucuronide, quercetin dirhamnoside, quercetin-*O*-pentoside and quercetin-*O*-deoxyhexoside, respectively. The same analogy was used to identify the isorhamnetin derivatives. Furthermore, strong absorption at 355 nm in the UV-Vis spectra, peaks 18 and 33 ([M-H]^−^ at *m*/*z* 477), and 29 ([M-H]^−^ at *m*/*z* 623) presented the fragment ion at *m*/*z* 315, characteristic of isorhamnetin derivatives, as the base peak in MS^2^ spectra and were identified as isorhamnetin hexoside isomers (loss of a hexosyl unit, −162 u) and isorhamnetin rutinoside (loss of a rutinoside, −308 u). Finally, the compound of peak 30 displayed retention time, and UV-Vis (λ_max_~350 nm) and MS spectra features ([M-H]^−^ at *m*/*z* 447 and [M-H-162]^−^ at *m*/*z* 285) similar to the available standard and was identified as kaempferol-3-*O*-glucoside.

#### 3.1.2. Quantification of Phenolic Compounds

Total phenolic compounds (TPC, computed as the sum of all individual compounds separated by HPLC) data showed that no chestnut by-product or extraction procedure consistently provided the highest or lowest polyphenol content or recovery (Table 2). Nonetheless, some broad conclusions can be drawn. Overall, leaves were the chestnut by-product with the highest TPC values, but that obtained with UAE-W did not differ from the other by-products (*p* > 0.05). Leaves’ hydroethanolic extracts obtained by maceration and UAE showed, respectively, the first and second-highest extraction yields in the amount of phenolic compounds, among the twelve extracts analysed (70.92 ± 2.72 and 53.97 ± 2.41 mg.g^−1^ dw). Moreover, the UAE-HE extract of burs (36.87 ± 1.09 mg.g^−1^ dw) and the UAE-W extract of shells (23.03 ± 0.26 mg.g^−1^ dw) presented the highest amount of TPC, within the extracts obtained from each one of these by-products. These findings support, in particular, the potential of UAE as a green extraction technique for the sustainable and efficient recovery of natural and bioactive molecules from agro-industrial material, where the hydroethanolic solvent displays good extraction efficiencies. Also, besides the first and second highest values of TPC, the MAE-W and UAE-W leaf extracts ranked fourth and fifth in polyphenol content (30.59 ± 1.2 and 20.49 ± 0.92 mg.g^−1^ dw) among all those analysed, corroborating the chestnut leaves as a rich, locally available and low-cost source of bioactive molecules. To the best of our knowledge, studies reporting the TPC of chestnut leaves as assessed by HPLC-DAD are limited to the study of Silva et al. (2020) [41], which also found this by-product to be the one with the highest content of total polyphenols in contrast to burs and shells. The TPC found by these authors for the conventional extract of leaves obtained with ethanol (28.5 ± 0.6 mg.g^−1^ dry extract) is within the range of absolute values of the present study but is lower than the content of three out of the four leaf extracts presented herein. Vella et al. (2018) [42] estimated the TPC of Italian chestnut by-products spectrophotometrically pointing out that the highest recovery of polyphenols was generally achieved from leaves, although the dependency on the extraction procedure. Their results showed that overall shells exhibited significantly lower levels of these compounds compared to leaves and burs, agreeing with our results as well.

Indeed, the TPC found in the composite sample containing inner and outer parts of chestnut shells were generally lower than the other by-products, varying from 23.03 ± 0.26 to 6.28 ± 0.09 mg.g^−1^ for the UAE-W and UAE-HE extracts (Table 2). Shells are the chestnut by-product whose polyphenol composition (profile and quantity) has been more thoroughly studied by HPLC-DAD and quite discrepant values have been reported. For instance, Pinto et al. (2021) [8] found a total of 15.85 mg.g^−1^ (dw) of polyphenols in aqueous MAE extract from Portuguese chestnut, comparable to those values found in the present study (9.69 ± 0.03 mg.g^−1^ of TPC for the same type of extract). However, this amount of polyphenols was achieved in 2 min at 85 °C, while in the former study, the total extraction procedure lasted 20 min, being 15 min at 75 °C under stirring [8]. In contrast, Cacciola et al. (2019) [28] quantified several phenolic compounds in aqueous MAE extract (60 min extraction at higher operating power) from an Italian *C. sativa*, leading to a total of 139.13 mg.g^−1^ dry extract, approximately 10 times higher than the quantities mentioned above, but involving large consumption of energy and resources. Interestingly, for shells, the best polyphenol yield was achieved with water, in accordance with results from Pinto et al. (2021) [8]. In that study, extraction efficiencies of shell extracts increased with the ethanol percentage increasing, but the extract yield did not reflect the polyphenols extracted, being related to the concomitant extraction of interfering substances [8]. Finally, chestnut burs were the by-product that presented, on average, the second highest content of phenolic compounds, ranging from 36.87 ± 1.09 mg.g^−1^ with the UAE-HE procedure to 9.56 ± 0.13 mg.g^−1^ with the MAE-W one, while MAC-HE and UAE-W presenting roughly half the highest total amount recovered for this by-product (21.85 ± 0.01 and 17.42 ± 0.16, respectively). These values greatly surpassed the ones reached by Silva et al. (2020) [41] for a conventional ethanolic extract of burs from the same chestnut variety (4.48 ± 0.01 mg.g^−1^ dry extract).

In this sense, apart from the extraction and analytical conditions addressed further in this section, it is important to note that the chemical composition of the plant material, especially concerning secondary metabolites such as phenolic compounds, may vary depending on their genetic variability, including species, cultivars and varieties to which they belong, as well as the specific plant part used. Even for the same species, edaphoclimatic conditions of cultivation, stage of maturity and post-harvest handling, among other factors, may differently impact the plant biosynthetic pathway leading to a differential accumulation of metabolites and consequently to a different chemical composition [10,63].

Individually, the phenolic compounds found in the highest concentrations in chestnut leaves were the bis-HHDP-glucose isomers for MAC-HE (13.40 ± 0.77 and 11.12 ± 0.77 mg.g^−1^ for isomers 1 and 2, chestanin isomer 1 for UAE-HE (10.80 ± 0.31 mg.g^−1^), and gallic acid for both UAE-W and MAE-W extracts (9.82 ± 0.25 and 10.86 ± 0.09 mg.g^−1^, respectively) (Table 2). Similarly, gallic acid (in the range of 4.28 in MAE-W to 17.55 mg.g^−1^ in MAC-HE), besides chestanin and bis-HHDP-glucose isomer (7.53 ± 0.31 and 5.67 ± 0.14 mg.g^−1^ in UAE-HE) also constituted the major compounds found in burs. In shells, the main constituents gallic (3.78 ± 0.06 to 14.84 ± 0.02 mg.g^−1^) and ellagic acids (up to 1.81 ± 0.04 mg.g^−1^ in MAC-HE) were followed by condensed tannins, namely procyanidin trimers (up to 1.35 ± 0.05 and 1.20 ± 0.04 in UAE-W), in contrast to the others by-products. When considering each type of extraction procedure, the concentration of gallic acid in extracts consistently exceeded that of the other individual polyphenols, except for the UAE-HE extracts of shells and burs, with higher or similar amounts of chestanin.

The constant and substantial presence of gallic acid and certain hydrolysable tannins in chestnut by-products is supported by the literature. This phenolic acid was found to be the major compound in shell extracts from Portuguese and Italian chestnuts, in ranges from 3.14 and 8.4 mg.g^−1^ dw [41,64] and 86.97 ± 1.32 to 150.09 ± 2.16 mg.g^−1^ dw [15], respectively. In addition, HHDP-NHTP-glucose isomer II was also one of the main components in shell extracts (3.10 mg.g^−1^ dw [64]). In the remaining few studies that present an individual quantification of phenolic compounds in chestnut by-products after chromatographic separation, cretanin and chestanin were found to be the major phenolic compounds in methanolic extracts of leaves and burs from “Marrone di Roccadaspide” chestnut, along with isorhamnetin-3-*O*-β-D-glucopyranoside [54]. Chestanin and ellagic acid were also found in high amounts in extracts of burs from another Italian cultivar, “Castagne di Montella”, which were obtained with different solvents [49].

### 3.2. Phenolic Compound Classes (Relative Percentage) in the Different Extracts

The relative contribution (percentage, %) of the different polyphenol classes to each extract may provide further insights into the composition of the different chestnut by-products, as well as into the feasibility of each extraction technique and solvent, under the conditions applied in this study, to recover specific phenolic compounds from them. In this sense, tannins, among condensed and hydrolysable, represented the most numerous class of phenolic compounds found in extracts of chestnut by-products in general, followed by flavonoids and phenolic acids (19, 13 and 7 representatives of the total 39 identified, respectively). Regarding the relative contribution to the total phenolic content of their respective samples, however, phenolic acids represented the major class of compounds in nine of the twelve extracts analysed, the remaining three presenting hydrolysable tannins as the most abundant one (Figure 4). In contrast to tannins and phenolic acids, flavonoids were not detected in all the extracts.

As indicated at the beginning of the section, shells presented the most distinct profile of phenolic compounds among the three chestnut by-products. Furthermore, such differences in terms of the individual compounds identified, extracts of this by-product also displayed the most diverse relative proportion of polyphenol classes in contrast to the other samples (Figure 4). Shells constituted the only chestnut by-product to present condensed tannins in its composition (accounting for 38, 54, 52 and 47% of total tannins (TT) for MAC-HE, UAE-HE, UAE-W and MAE-W extracts, respectively), and overall lower relative contribution of total hydrolysable tannins (THT) to TPC than leaf and bur extracts. In addition, shell extracts were characterized by the low or null presence of total flavonoids (TF), represented only by the flavan-3-ol monomer of condensed tannins detected in the MAE-W (5%), but primarily by having the largest share of their polyphenolic made up of phenolic acids (TPA) that accounted for 50 to 85% of the total. These results aligned with earlier research that found phenolic acids as the primary constituents or nearly all the polyphenols in chestnut shells [28,41,65]. Lameirão et al. (2020) [65] and Pinto et al. (2021) [8] also reported the presence of condensed tannins and their flavan-3-ol monomers, besides low amounts of other flavonoids in aqueous extracts of shells; however, whereas the former did not detect hydrolysable tannins in the extract, in the latter these compounds comprised 65% of the TPC. In general, the relative percentage of phenolic classes was comparable among the four extracts of chestnut shells, regardless of the extraction method and solvent. A closer look, though, shows that maceration with 80% ethanol resulted in a greater extraction of phenolic acids and a reduced recovery of total tannins.

In contrast to their counterparts extracted with the same method and solvent, bur extracts presented lower TPA and higher TF than shells, and lower TF and higher TPA than the leaves, in relative percentages. Moreover, bur extracts generally had a similar proportion of total tannins than their shell and leaf counterparts, except for the MAC-HE extract, three-fold inferior to the leaf sample. The proportion among the polyphenol classes through the four bur extracts was not as consistent as that noticed in shells. The relative contribution of the different classes to the TPC content within these extracts was more dependent on the extraction method. For instance, phenolic acids were the main class of compounds found in MAC-HE, UAE-W and MAE-W extracts of burs. Within these three extracts, whereas the magnitude of the relative contribution of the phenolic classes followed the same order (phenolic acids > hydrolysable tannins ≥ flavonoids), a large difference in their ratio could be noticed (approximately 12:2:1, 5:2:1 and 2:1:1 for the respective classes in MAC-HE, UAE-W and MAE-W). This also indicates, as observed in shell extracts, that phenolic acids were more effectively extracted by maceration with 80% ethanol from burs. The UAE-HE extract, on the other hand, contained 54% THT, 21% TF and 25% TPA, differing from the other extracts obtained using either the same method but a different solvent or the same solvent but another method.

Leaves presented the highest proportion of TF among the chestnut by-products, this is in line with the findings of Silva et al. (2020) [41]. The percentage of this class was generally consistent throughout all four extracts (ranging from 28 to 37% of the total). As a result, leaf extracts presented the most uniform distribution of the three phenolic classes to their profiles among the chestnut by-products, most evident in the MAE-W extract with 37, 33 and 30% of TPA, TF, and THT, respectively. Nonetheless, the proportion among the phenolic classes were more dependent on the extraction solvent, with the pairs MAC-HE and UAE-HE, and UAE-W and MAE-W, sharing a greater degree of similarity. In this regard, the extracts obtained with water presented phenolic acids as the primary contributors to their TPC, whereas the hydroethanolic extracts contained a higher percentage of hydrolysable tannins. Notably, leaf extracts obtained with 80% ethanol presented one of the highest concentrations of hydrolysable tannins and the lowest percentage of phenolic acids among the twelve extracts analysed.

Therefore, still in relative proportions of phenolic classes but focusing on the extraction technique and solvent, we can further build some generalizations from the above. Maceration with 80% ethanol was proportionally more efficient in recovering phenolic acids in shells and burs, while the opposite was noticed for the same extract of leaves. This latter followed the same pattern noticed for the other group of hydroethanolic extracts obtained by UAE. The UAE-HE extracts overall had the highest proportions of tannins against phenolic acids within the extracts of the same by-product, even when the TPA was still the primary class. As mentioned before in the individual quantitation, this was the only extraction procedure in which the content of chestanin extracted surpassed that of gallic acid. Therefore, with 80% EtOH, UAE extracted higher proportions of tannins and also flavonoids than the conventional solid–liquid extraction (MAC). When the solvent was replaced by water, UAE extracts of shells kept almost the same profile, while for the other two, the extraction of phenolic acids increased in a way that they presented around 50–60% of TPA. Moreover, all UAE-W extracts presented approximately 25–30% of TT. Also, with water, extracts obtained by MAE also presented TPA as the main class and overall showed the less heterogeneous relative proportion of phenolic classes among all the extraction techniques.

### 3.3. Phenolic Compound Classes and Total Phenolic Compounds (Absolute Values) in the Different Extracts

When considering not the relative percentage but the absolute values of compounds quantified in each class (mg.g^−1^ freeze-dried extracts), hydrolysable tannins represented the most important group of polyphenols of the chestnut extracts quantitatively, corresponding to 39.44 ± 1.91 and 26.23 ± 1.19, and 19.82 ± 0.77 mg.g^−1^ in MAC-HE and UAE-HE extracts of leaves and UAE-HE extract of burs, respectively (Figure 5, Table 1). Even though these compounds were the main class of polyphenols in only three of the twelve extracts, these extracts were precisely the first three in terms of polyphenol levels. Then, flavonoid levels up to 22.78 ± 0.77 mg.g^−1^ were detected in leaves, while phenolic acids’ levels up to 17.78 ± 0.06 mg.g^−1^ were found in burs, both in MAC-HE extracts.

For leaves, the aqueous extracts showed a much lower recovery of tannins and flavonoids (only 4.9–9.06 mg.g^−1^ and 5.77–10.12 mg.g^−1^ for a maximum of 39.44 and 22.78 mg.g^−1^ recovered using a hydroethanolic solution, respectively) and slightly more phenolic acids. For burs, the highest recoveries of TPA, as well as TF and THT, were also registered for hydroethanolic extracts obtained with MAC and UAE, respectively. Only for shells, the highest absolute recoveries of phenolic classes were found for the UAE-W extract. They constitute the only by-product in which the UAE extract obtained with water exceeded the TPC of its counterpart obtained with 80% ethanol solution. In their turn, leaves were the only by-product in which the absolute TPC of MAE-W exceeded that of UAE-W, whereas only in burs the UAE-HE presented the highest amount of phenolics than the MAC-HE extract.

The above information is summarised and visualised in the PCA performed with quantitative data (absolute values, mg.g^−1^) of total phenolic compounds and polyphenolic classes (Figure 5B). The first two components explained most of the original variance (85.2%). PC1 explained around 63% of the total variance and overall showed the spatial separation of the extracts from shells on the negative side and most of the extracts from leaves on the positive side. The first principal component had a greater association with the total content of flavonoids, hydrolysable tannins and phenolic compounds (first quadrant), particularly high in hydroethanolic extracts of leaves, and total condensed tannins (second quadrant), class of polyphenols that discriminate shells from the other chestnut by-products. In this sense, on the positive side of PC1 were found three extracts from leaves and one from burs, being the higher the TPC, THT and TF, the closer the extract is located to the positive extremity. Within them, three were extracted with 80% ethanol, two obtained with UAE and the remaining one with maceration, besides one extract obtained with MAE and water. PC2 explained 22% of data variance and was positively related to total phenolic acid levels. This component showed the spatial separation of the UAE-W extract from shells and the MAC-HE extracts from shells and burs, characterised by higher contents of TPA in the positive side of the component (second quadrant), distant from MAE-W extracts from burs and shells and UAE-HE extract from shells (third quadrant) at the opposite side in PC2.

### 3.4. Bioactivities

#### 3.4.1. Antioxidant Capacity

In the field of Food Science and Technology, antioxidants are expected to extend the useful shelf-life of food and beverages and prevent the deterioration of their components by inhibiting, controlling or delaying oxidation reactions. The screening on the antioxidant potential of chestnut by-products’ extracts as assessed by different colourimetric assays (TBARS, DPPH and RP) is presented in Table 3. Results were presented as EC_50_ (extract concentration capable of exerting 50% of antioxidant capacity) rather than equivalents of a given standard to facilitate the selection of extracts’ effective concentrations for any eventual incorporation into a formulation.

With regard to TBARS, which measures the capacity of the extracts to inhibit the formation of thiobarbituric acid reactive substances, secondary products of lipid oxidation, the lowest EC_50_ values were found for the bur extracts, irrespective of the extraction procedure (from 0.002 ± 0.0001 to 0.008 ± 0.0002 mg.mL^−1^). Stated alternatively, smaller amounts of bur extracts, in contrast to others, were sufficient to inhibit lipidic peroxidation, thereby presenting a higher antioxidant efficiency. For instance, the antioxidant capacity of the UAE-HE extract of burs was roughly 100 and 1000 times higher than its respective counterparts from shells and leaves. It is worth noting that bur extracts were equally or even more efficient against peroxidation than the antioxidant Trolox used as a positive control. Although these extracts were not the richest ones in TPC among all those analysed, when comparing only the four bur extracts, those prepared with 80% EtOH and exhibiting the highest antioxidant capacity had, on average, higher TPC and TPA than the aqueous extracts. Conversely, for shells and leaves, the extracts obtained with water either by UAE or MAE generally presented the best antioxidant capacities, especially the UAE-W extract of shells (0.018 ± 0.0005 mg.mL^−1^) and MAE-W extract of leaves (0.08 ± 0.02 mg.mL^−1^). Whereas the former exhibited the highest TPC among shell extracts, the latter was only the third leaf extract in terms of polyphenol content, but both presented the highest TPA levels of their respective groups. On the other hand, the UAE-HE extract of leaves, which ranked second in terms of TPC, had the highest EC_50_ value (2.0 ± 0.2 mg.mL^−1^) and hence the lowest antioxidant capacity among the extracts studied. Therefore, the antioxidant capacity in this assay did not increase the TPC.

In contrast to TBARS, shell extracts generally presented similar or better antioxidant responses in both DPPH and RP assays than other by-products’ extracts. The best antioxidant capacities, as measured by the respective abilities to scavenge the DPPH^•^ radical and reduce Fe^3+^ to Fe^2+^ were found for the MAE-W and MAC-HE extract of shells (EC_50_ of 0.07 ± 0.01 mg.mL^−1^ for both of them). Conversely, another shell extract, the one obtained by UAE-W that displayed good activity in TBARS, presented the lowest antioxidant potential in both DPPH and RP methods, along with its UAE-W counterpart from burs in the RP assay. Overall, the other extracts presented scavenging capacity and reducing power about five to ten times lower than positive controls, except for the MAC-HE of leaves extract for the latter. Barreira et al. (2008; 2010) [39,40] evaluated the antioxidant capacity of aqueous extracts of different chestnut parts, which do not include burs, and also reported the superior antioxidant potential of shells compared to other samples as measured by a number of methodologies. The study of Silva et al. (2020) [41], however, found that the ethanolic extract of chestnut leaves was more efficient than those of burs and shells in scavenging the DPPH^•^ radical (EC_50_ 0.03 mg.mL^−1^).

Together these results suggest the overall lack of correlation between the total phenolic content and the antioxidant capacity of the extracts. This may be due to various factors such as the potency of individual phenolic compounds with different active groups, eventual interactions among the phenolic compounds or other plant metabolites concomitantly extracted and possibly all these events occurring simultaneously. The complexity of crude plant extracts makes it challenging to predict the net effect of synergistic, additive and antagonistic interactions among the compounds, but it is likely that such interactions play a role in their overall antioxidant capacity [66].

#### 3.4.2. Hepatotoxicity

The extracts were tested for cytotoxicity in human non-tumour culture (PLP2), the results being shown in Table 3. The non-hepatotoxic nature of shell and burs extracts was observed at the maximum tested concentration of 400 mg.mL^−1^ using UAE-HE and UAE-W extraction methods as also using MAC-HE. These findings suggest that both extraction methods yield extracts with similar results, indicating their effectiveness in extracting valuable bioactive compounds while not reaching hepatotoxicity levels.

On the other hand, the extracts obtained by MAE revealed GI_50_ values, indicating possible toxicity in PLP2 cells. Nevertheless, these values are much higher than the ones revealed by the positive control ellipticine, indicating significant lower toxicity for PLP2 cells than the commonly applied drug. These findings suggest preferably the application of MAC and UAE extractions.

#### 3.4.3. Antimicrobial Activity

Another valuable property that has been associated with phenolic compounds is their antimicrobial activity. In this sense, the antibacterial and antifungal activities of *C. sativa* extracts were assessed against several microbial strains of food relevance, as this screening prospects their use as high-value-added ingredients, primarily as food preservatives. The results expressed as minimum inhibitory concentration (MIC), minimum bactericidal concentration (MBC) and fungicidal concentration (MFC) are shown in Table 4.

The MAE-W extract of burs showed one of the most promising antimicrobial results among the extracts analysed as was able to cause the death of *E. cloacae*, *P. aeruginosa* and *S. aureus* at an MBC of 5 mg.mL^−1^, as well as the *S. enterocolitica* at the maximum concentration tested (10 mg.mL^−1^). Additionally, this extract demonstrated bacteriostatic activity against *L. monocytogenes* and *S. aureus* at concentrations as low as 0.6 mg.mL^−1^ and 0.3 mg.mL^−1^, respectively. The important antibacterial activity of bur MAE-W extract was further noticed against *P. aeruginosa* (MIC/MBC 2.5/5 mg.mL^−1^), which was not eliminated by any other chestnut extract at the maximum concentration tested. The UAE-W extract of burs also presented bactericidal and bacteriostatic activities against *S. aureus* at the respective concentrations of 10 and 0.6 mg.mL^−1^, and along with their UAE-W counterparts of shells and leaves, constitute the only extracts capable of suppressing the growth of *E. coli* among the twelve evaluated in concentrations lower than 10 mg.mL^−1^. Moreover, the MAE-W extract of leaves also presented good bacteriostatic activity, especially against *S. aureus* (MIC 0.3 mg.mL^−1^). Of note, the MIC of MAE-W extracts of leaves and burs against *S. aureus* were only two times that of the positive control ampicillin (0.15 mg.mL^−1^). These results indicate that extracts with high antimicrobial efficiency can be obtained by using innovative green methods and water, the greenest solvent option, particularly from chestnut burs. Additionally, irrespective of the method or solvent used, leaf extracts inhibited the growth of all bacteria tested at concentrations of at least 10 mg.mL^−1^.

For shells, the hydroethanolic extracts presented greater antimicrobial potential than the aqueous ones. This agrees in parts with Genovese et al. (2021) [32], who prepared conventional extracts of chestnut inner shells and found that the ethanolic extract, not the aqueous, presented inhibitory activity against the several strains tested. In the present study, lower concentrations of both MAC-HE and UAE-HE were similarly efficient against *S. aureus* (MIC 0.6 mg.mL^−1^), *L. monocytogenes* (MIC 1.25 mg.mL^−1^) and *Y. enterocolitica* (MIC 1.25 mg.mL^−1^). In addition, compared to the hydroethanolic extracts of burs and leaves, shell extracts showed equal or superior inhibitory activity against the bacteria strains analysed.

The Gram-positive bacteria were generally more sensitive to chestnut by-products’ extracts than the Gram-negative bacteria, particularly *S. aureus* against which the lower MIC values were found. Nonetheless, more records of the lowest bactericidal concentration registered (MBC of 5 mg.mL^−1^) were found against Gram-negative bacteria. Several authors have indeed reported that the cell wall characteristics of Gram-negative bacteria pose a particular challenge against the access to the membrane or entry of phenolic compounds into the cell cytoplasm, but others have concluded that the activity is strain-dependent [67]. To the best of our knowledge, this is the first study to report bactericidal concentrations of chestnut bur, leaf and shell extracts. Finally, no extract demonstrated the ability to cause the death of *A. brasiliensis* and *A. fumigatus*, while bur extracts inhibited their growth at 10 mg.mL^−1^, regardless of the extraction method. Fernández-Agulló et al. (2014) [68] also reported the absence of antifungal activity for chestnut bur and shell extracts.

## 4. Conclusions

In conclusion, chestnut by-products, particularly leaves, offer a valuable source of phenolic compounds for various applications. The polyphenol distribution varied among different parts of the chestnut plant, with phenolic acids and hydrolysable tannins being the predominant compounds. The choice of extraction method and solvent influenced the recovery of polyphenols, UAE-HE shows effective results. Bur and shell extracts exhibited higher antioxidant capacity than leaf extracts, while aqueous extracts of burs and leaves showed promising antimicrobial activity. These findings contribute to understanding the polyphenolic composition of chestnut extracts and their potential applications in the food and nutraceutical industries. Chestnut by-products have the potential to be a sustainable source of natural antioxidant and antimicrobial agents. Future research should focus on optimizing extraction methods and exploring the recovery of bound polyphenols from burs and shells to maximize the utilization of these agricultural waste materials.

## Figures and Tables

**Figure 1 foods-12-02596-f001:**
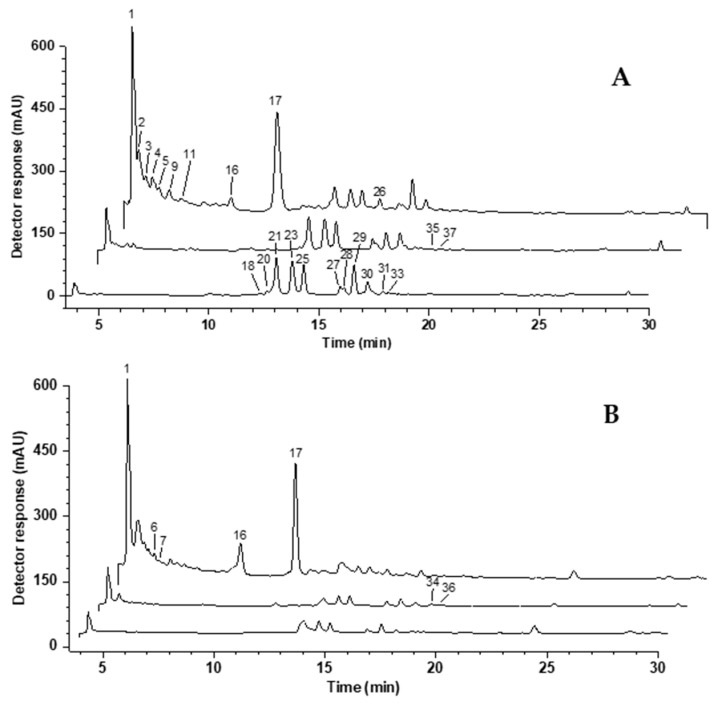
Representative chromatograms, obtained by HPLC-DAD, of phenolic compound extracts from chestnut by-products obtained by ultrasound-assisted extraction with 80% ethanol (UAE-HE). (**A**) leaf extract; (**B**) bur extract; (**C**) shell extract. The chromatograms were processed at 280, 330 and 370 nm and are displayed in this order. Peak identification can be found in Table 1.

**Figure 2 foods-12-02596-f002:**
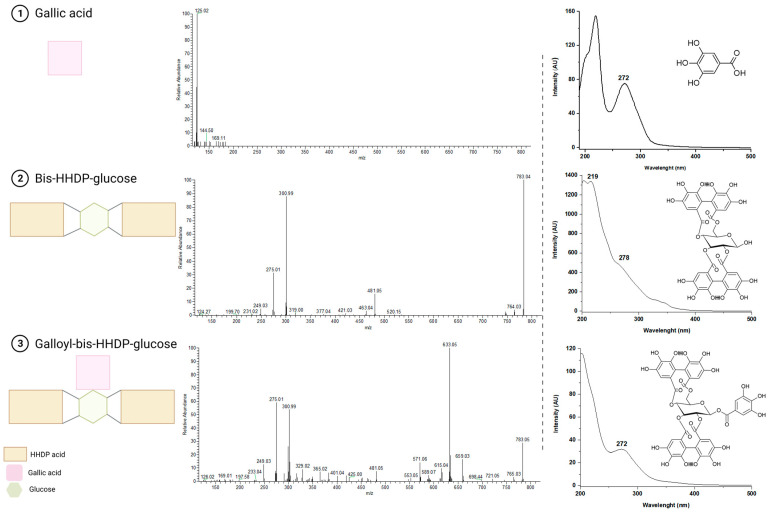
Schematic representation along with UV-Vis, MS/MS spectra and chemical structures of gallic acid, bis-HHDP-glucose and galloyl-bis-HHDP-glucose. Created with BioRender.com (accessed on 5 April 2023).

**Figure 3 foods-12-02596-f003:**
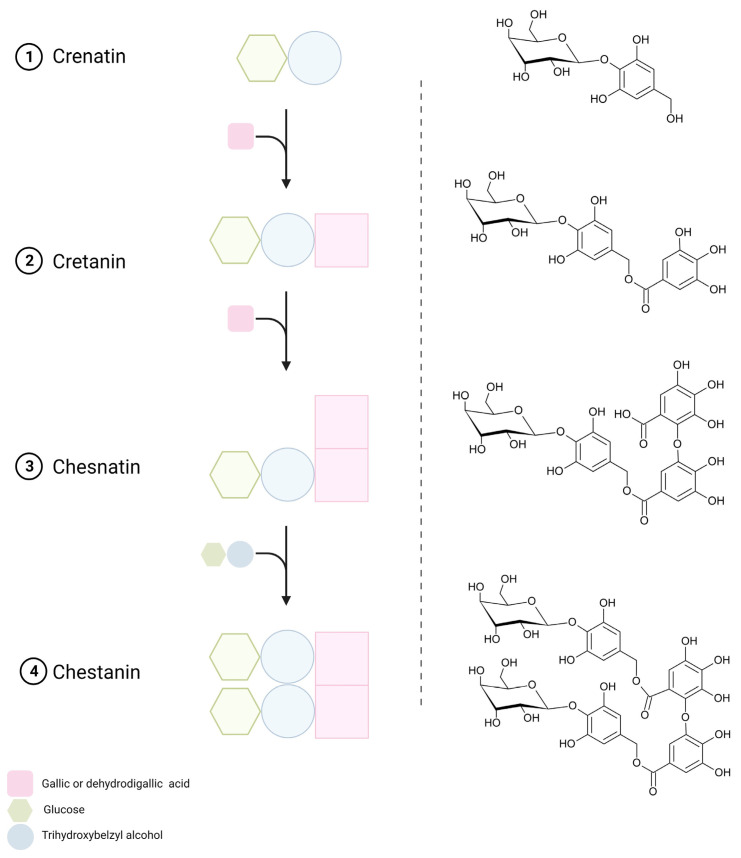
Schematic representation of the different groups composing of the molecules of gallotannins and their chemical structures. Cretanin might be regarded as a monoester of gallic acid with crenatin, while chesnatin and chestanin can be a mono- and a diester of dehydrodigallic acid with one or two molecules of crenatin, respectively [52]. Created with BioRender.com (accessed on 5 April 2023).

**Figure 4 foods-12-02596-f004:**
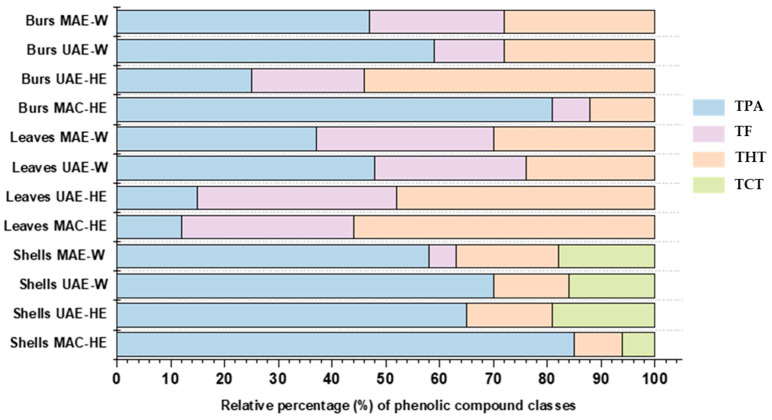
Relative contribution (percentage) of the different classes of phenolic compounds from the total content of these compounds found in the twelve extracts of chestnut by-products. TPA: Total phenolic acids, TF: Total flavonoids; THT: Total hydrolysable tannins; TCT: Total condensed tannins; MAC-HE: maceration carried out with hydroethanolic solvent (80% EtOH); UAE-HE: Ultrasound-Assisted Extraction carried out with hydroethanolic solvent (80% EtOH); UAE-W: Ultrasound-Assisted Extraction carried out with water; MAE-W: Microwave-Assisted Extraction carried out with water.

**Figure 5 foods-12-02596-f005:**
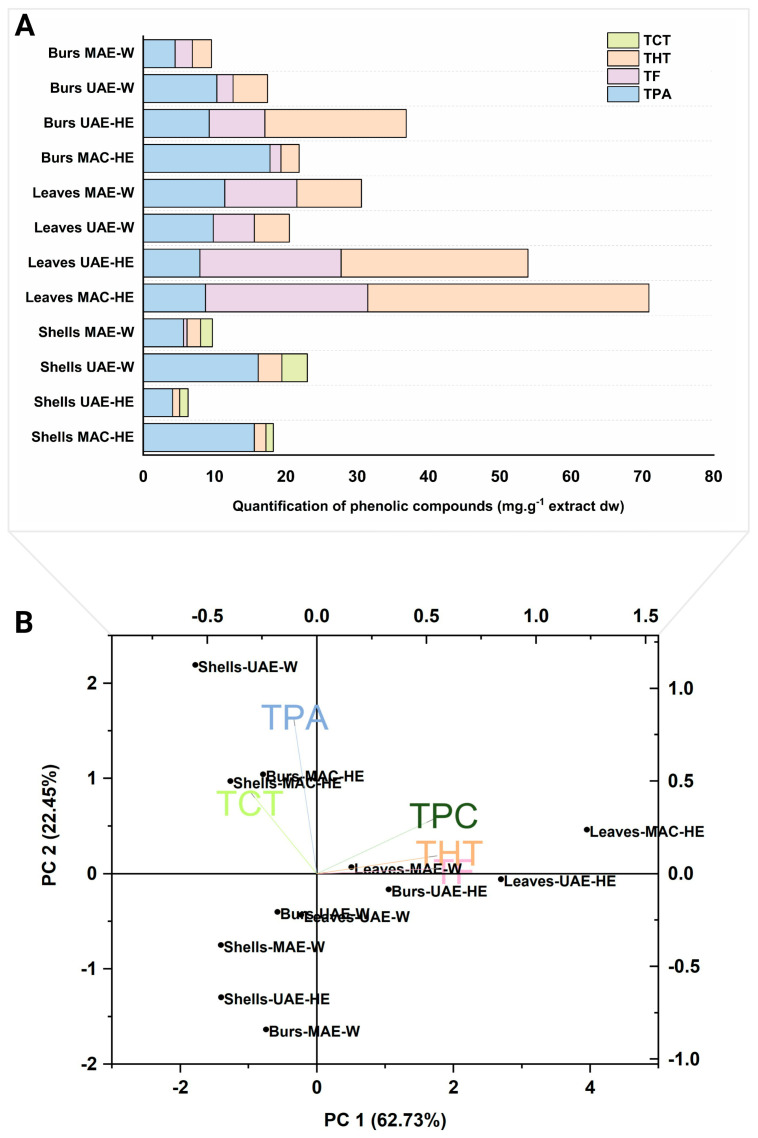
(**A**) Total phenolic compound content (mg.g^−1^ freeze-dried extract) in different extracts of chestnut by-products as occurring in the classes of these compounds. (**B**) Principal Component Analysis (PCA) of the quantitative data of phenolic compounds of the chestnut by-product extracts.

**Table 1 foods-12-02596-t001:** Chromatographic and spectroscopy characteristics of phenolic compounds in different extracts from *C. sativa* by-products.

Peaks ^a^	Tentative Identification	Rt (min) ^b^	Λmax (nm) ^c^	[M-H]^−^ (*m*/*z*)	MS^2^ Fragments (*m*/*z*) ^d^	Phenolic Compound Class	Chestnut By-Products		
							L	S	B
1	Gallic acid	3.9	270	169	125 (100)	Phenolic acid			
2	Bis-HHDP-glucose (pedunculagin I) (isomer 1)	4.38	278	783	481 (15), 301 (87)	Hydrolysable tannin			
3	Galloyl-bis-HHDP-glucose (isomer 1)	4.76	288	935	633 (100), 301 (18)	Hydrolysable tannin			
4	Chesnatin (isomer 1)	4.91	270	637	467 (100), 305 (23)	Hydrolysable tannin			
5	Digalloyl-HHDP-glucose (isomer 1)	5.09	280	785	633 (100), 483 (24), 301 (38)	Hydrolysable tannin			
6	Trigalloyl-HHDP-glucose	5.11	274	937	937 (100), 637 (20), 301 (12)	Hydrolysable tannin			
7	Galloyl-bis-HHDP-glucose (isomer 2)	5.45	275	935	633 (100), 301 (18)	Hydrolysable tannin			
8	Digalloyl-HHDP-glucose (isomer 1)	5.46	280	785	633 (100), 483 (24), 301 (38)	Hydrolysable tannin			
9	Bis-HHDP-glucose (pedunculagin I) (isomer 2)	5.56	278	783	481 (13), 301 (85)	Hydrolysable tannin			
10	Procyanidin tetramer	5.92	281	1153	865 (22), 713 (4), 577 (33), 575 (16), 561 (20), 289 (100)	Condensed tannin			
11	Chesnatin (isomer 2)	5.95	272	637	467 (100), 305 (23)	Hydrolysable tannin			
12	(-)-Epicatechin	6.21	280	289	245 (100)	Flavonoid			
13	Procyanidin trimer	6.55	281	865	451 (44), 425 (59), 407 (97), 289 (65)	Condensed tannin			
14	Galloyl-bis-HHDP-glucose (isomer 3)	6.92	281	935	633 (100), 301 (18)	Hydrolysable tannin			
15	Galloyl-bis-HHDP-glucose (isomer 4)	8.22	281	935	633 (100), 301 (18)	Hydrolysable tannin			
16	Cretanin	8.38	274	469	169 (100)	Hydrolysable tannin			
17	Chestanin (isomer 1)	10.47	274	937	637 (6), 467 (100), 305 (7), 169 (17)	Hydrolysable tannin			
18	Isorhamnetin-*O*-hexoside	12.34	353	477	315 (100)	Glycosylated flavonoid			
19	Ellagic acid	12.53	366	301	135 (100)	Phenolic acid			
20	Quercetin-3-*O*-rutinoside	12.67	347	609	301 (100)	Glycosylated flavonoid			
21	Quercetin-deoxyhexosyl-hexoside	13.08	346	609	301 (100)	Glycosylated flavonoid			
22	Galloyl-bis-HHDP-glucose (isomer 5)	13.27	281	935	633 (100), 301 (18)	Hydrolysable tannin			
23	Quercetin-3-*O*-glucuronide	13.77	352	477	301 (100)	Glycosylated flavonoid			
24	3-*O*-galloylpunicalin	14.21	288	933	753 (45), 631 (26), 597 (18), 451 (39), 425 (11), 301 (100)	Hydrolysable tannin			
25	Quercetin-*O*-hexoside	14.71	354	463	301 (100)	Glycosylated flavonoid			
26	Chestanin (isomer 2)	15.12	274	937	637 (6), 467 (100), 305 (7), 169 (17)	Hydrolysable tannin			
27	Quercetin dirhamnoside	15.95	334	593	301 (100)	Glycosylated flavonoid			
28	Quercetin-*O*-pentoside	16.14	345	433	301 (100)	Glycosylated flavonoid			
29	Isorhamnetin-3-*O*-rutinoside	16.62	355	623	315 (100)	Glycosylated flavonoid			
30	Kaempferol-3-*O*-glucoside	17.23	348	447	285 (100)	Glycosylated flavonoid			
31	Quercetin-*O*-deoxyhexoside	17.87	349	447	301 (100)	Glycosylated flavonoid			
32	Procyanidin trimer	18.02	280	867	715 (46), 577 (100), 409 (58), 287 (63)	Condensed tannin			
33	Isorhamnetin-*O*-hexoside (isomer 1)	18.17	353	477	315 (100)	Glycosylated flavonoid			
34	Methyl ellagic acid hexoside	18.46	362	477	301 (100)	Glycosylate Phenolic acid			
35	Ellagic acid trimethyl-glucoside (isomer 1)	18.6	355	551	343 (100)	Glycosylate Phenolic acid			
36	Isorhamnetin-*O*-hexoside (isomer 2)	18.75	353	477	315 (100)	Glycosylated flavonoid			
37	Methyl ellagic acid deoxyhexoside (isomer 1)	18.88	368	461	315 (100), 301 (32)	Glycosylate Phenolic acid			
38	Methyl ellagic acid deoxyhexoside (isomer 2)	19.89	367	461	315 (100), 301 (32)	Glycosylate Phenolic acid			
39	Ellagic acid trimethyl-glucoside (isomer 2)	23.97	355	551	343 (100)	Glycosylate Phenolic acid			

^a^ Peaks numbered according to the chromatogram shown in Figure 1. ^b^ Retention time on C_18_ column. ^c^ Gradient of 0.1% formic acid and acetonitrile. ^d^ MS/MS fragments are followed by their relative abundance in the spectrum in parentheses. Grey cells in the L, S and B columns indicate the detection of the compound in chestnut leaves, shells, and burs.

**Table 2 foods-12-02596-t002:** Quantification of phenolic compounds in different extracts from *C. sativa* by-products.

Phenolic Compound ^1^	Individual Quantification (mg.g^−1^) ^2^											
	Leaves				Shells				Burs			
	MAC-HE	UAE-HE	UAE-W	MAE-W	MAC-HE	UAE-HE	UAE-W	MAE-W	MAC-HE	UAE-HE	UAE-W	MAE-W
Gallic acid	7.91 ± 0.010	7.24 ± 0.17	9.82 ± 0.25	10.86 ± 0.092	13.75 ± 0.075	3.78 ± 0.056	14.84 ± 0.02	4.89 ± 0.062	17.55 ± 0.067	7.58 ± 0.012	10.31 ± 0.093	4.28 ± 0.03
Bis-HHDP-glucose (pedunculagin I) (isomer 1)	13.4 ± 0.77	2.39 ± 0.15	nd	nd	nd	nd	nd	nd	2.04 ± 0.02	5.67 ± 0.14	2.23 ± 0.03	0.5 ± 0.02
Galloyl-bis-HHDP-glucose (isomer 1)	2.4 ± 0.11	3.77 ± 0.2	nd	nd	0.40 ± 0.01	0.23 ± 0.002	0.90 ± 0.02	0.44 ± 0.01	nd	nd	nd	nd
Chesnatin (isomer 1)	nd	nd	2.04 ± 0.13	2.93 ± 0.12	nd	nd	nd	nd	nd	nd	nd	nd
Digalloyl-HHDP-glucose (isomer 1)	nd	nd	nd	nd	0.32 ± 0.02	0.196 ± 0.001	0.49 ± 0.02	0.31 ± 0.01	nd	nd	nd	nd
Trigalloyl-HHDP-glucose	nd	nd	nd	nd	nd	nd	nd	nd	0.18 ± 0.003	1.26 ± 0.05	1.55 ± 0.05	0.47 ± 0.02
Galloyl-bis-HHDP-glucose (isomer 2)	nd	nd	nd	nd	nd	nd	nd	nd	0.065 ± 0.004	1.71 ± 0.11	nd	0.53 ± 0.03
Digalloyl-HHDP-glucose (isomer 1)	nd	nd	nd	nd	0.194 ± 0.001	0.119 ± 0.001	0.33 ± 0.01	0.178 ± 0.001	nd	nd	nd	nd
Bis-HHDP-glucose (pedunculagin I) (isomer 2)	11.12 ± 0.31	2.87 ± 0.12	nd	nd	nd	nd	nd	nd	nd	nd	nd	nd
Procyanidin tetramer	nd	nd	nd	nd	nd	0.45 ± 0.01	1.01 ± 0.04	0.52 ± 0.002	nd	nd	nd	nd
Chesnatin (isomer 2)	2.01 ± 0.09	2.89 ± 0.14	1.19 ± 0.09	2.02 ± 0.14	nd	nd	nd	nd	nd	nd	nd	nd
(-)-Epicatechin	nd	nd	nd	nd	nd	nd	nd	0.50 ± 0.004	nd	nd	nd	nd
Procyanidin trimer	nd	nd	nd	nd	0.59 ± 0.01	0.426 ± 0.004	1.35 ± 0.05	0.62 ± 0.005	nd	nd	nd	nd
Galloyl-bis-HHDP-glucose (isomer 3)	nd	nd	nd	nd	nd	0.193 ± 0.001	0.59 ± 0.01	0.21 ± 0.001	nd	nd	nd	nd
Galloyl-bis-HHDP-glucose (isomer 4)	nd	nd	nd	nd	0.28 ± 0.01	0.152 ± 0.001	0.71 ± 0.01	0.21 ± 0.002	nd	nd	nd	nd
Cretanin	1.84 ± 0.12	1.87 ± 0.13	1.01 ± 0.05	1.28 ± 0.1	nd	nd	nd	nd	0.186 ± 0.002	3.66 ± 0.17	0.79 ± 0.03	0.52 ± 0.02
Chestanin (isomer 1)	7.47 ± 0.43	10.8 ± 0.31	0.66 ± 0.02	2.82 ± 0.15	nd	nd	nd	nd	0.077 ± 0.004	7.53 ± 0.31	0.26 ± 0	0.65 ± 0.03
Isorhamnetin-*O*-hexoside	0.38 ± 0.04	0.41 ± 0.02	0.3 ± 0.05	0.27 ± 0.04	nd	nd	nd	nd	nd	nd	nd	0.14 ± 0.01
Ellagic acid	nd	nd	nd	nd	1.81 ± 0.04	0.33 ± 0.003	1.27 ± 0.02	0.74 ± 0.03	nd	nd	nd	nd
Quercetin-3-*O*-rutinoside	0.33 ± 0.01	0.58 ± 0.09	1.54 ± 0.07	0.41 ± 0.02	nd	nd	nd	nd	0.18 ± 0.004	2.67 ± 0.12	1.24 ± 0.07	0.61 ± 0.02
Quercetin-deoxyhexosyl-hexoside	4.09 ± 0.17	4.62 ± 0.38	0.92 ± 0.06	1.87 ± 0.15	nd	nd	nd	nd	nd	nd	nd	nd
Galloyl-bis-HHDP-glucose (isomer 5)	nd	nd	nd	nd	0.47 ± 0.02	nd	nd	0.29 ± 0.01	nd	nd	nd	nd
Quercetin-3-*O*-glucuronide	3.49 ± 0.02	4.23 ± 0.32	1.12 ± 0.05	2.61 ± 0.13	nd	nd	nd	nd	0.36 ± 0.004	1.66 ± 0.15	0.75 ± 0.02	0.57 ± 0.03
3-*O*-galloylpunicalin	nd	nd	nd	nd	nd	0.10 ± 0.002	0.29 ± 0.01	0.24 ± 0.001	nd	nd	nd	nd
Quercetin-*O*-hexoside	2.68 ± 0.1	3.03 ± 0.12	0.25 ± 0.04	1.29 ± 0.15	nd	nd	nd	nd	0.063 ± 0.001	1.18 ± 0.05	nd	0.4 ± 0.01
Chestanin (isomer 2)	1.21 ± 0.08	1.65 ± 0.14	nd	nd	nd	nd	nd	nd	nd	nd	nd	nd
Quercetin dirhamnoside	0.62 ± 0.07	0.62 ± 0.05	0.33 ± 0.01	0.93 ± 0.1	nd	nd	nd	nd	0.62 ± 0.004	0.41 ± 0.05	nd	0.16 ± 0.01
Quercetin-*O*-pentoside	0.8 ± 0.01	0.85 ± 0.04	nd	nd	nd	nd	nd	nd	nd	nd	nd	nd
Isorhamnetin-3-*O*-rutinoside	2.63 ± 0.17	3.08 ± 0.2	0.93 ± 0.05	1.33 ± 0.05	nd	nd	nd	nd	0.10 ± 0.005	0.83 ± 0.07	0.29 ± 0.04	0.27 ± 0.01
Kaempferol-3-*O*-glucoside	6.98 ± 0.1	1.51 ± 0.09	0.39 ± 0.04	0.89 ± 0.1	nd	nd	nd	nd	0.1 ± 0.002	0.41 ± 0.01	nd	0.15 ± 0.01
Quercetin-*O*-deoxyhexoside	0.48 ± 0.05	0.49 ± 0.04	nd	0.28 ± 0.04	nd	nd	nd	nd	nd	0.404	nd	0.123
Procyanidin trimer	nd	nd	nd	nd	0.43 ± 0.005	0.29 ± 0.005	1.20 ± 0.04	0.52 ± 0.003	nd	nd	nd	nd
Isorhamnetin-*O*-hexoside (isomer 1)	0.31 ± 0.01	0.4 ± 0.02	nd	0.25 ± 0.01	nd	nd	nd	nd	nd	nd	nd	nd
Methyl ellagic acid hexoside	nd	nd	nd	nd	nd	nd	nd	nd	0.11 ± 0.004	0.4 ± 0.01	nd	0.12 ± 0.01
Ellagic acid trimethyl-glucoside (isomer 1)	0.34 ± 0.001	0.33 ± 0.01	nd	nd	nd	nd	nd	nd	nd	nd	nd	nd
Isorhamnetin-*O*-hexoside (isomer 2)	nd	nd	nd	nd	nd	nd	nd	nd	0.10 ± 0.004	0.23 ± 0.04	nd	nd
Methyl ellagic acid deoxyhexoside (isomer 1)	0.45 ± 0.02	0.35 ± 0.02	nd	0.56 ± 0.02	nd	nd	nd	nd	0.11 ± 0.003	0.32 ± 0.01	nd	0.17 ± 0
Methyl ellagic acid deoxyhexoside (isomer 2)	nd	nd	nd	nd	nd	nd	nd	nd	nd	0.34 ± 0.01	nd	nd
Ellagic acid trimethyl glucoside (isomer 2)	nd	nd	nd	nd	nd	nd	nd	nd	nd	1.01 ± 0.02	nd	nd
	**Total quantification (mg.g^−1^) ^2^**											
	**Leaves**				**Shells**				**Burs**			
	**MAC-HE**	**UAE-HE**	**UAE-W**	**MAE-W**	**MAC-HE**	**UAE-HE**	**UAE-W**	**MAE-W**	**MAC-HE**	**UAE-HE**	**UAE-W**	**MAE-W**
**Total phenolic acids** **(TPA)**	8.71 ± 0.04	7.93 ± 0.15	9.82 ± 0.25	11.42 ± 0.07	15.56 ± 0.04	4.11 ± 0.06	16.11 ± 0.04	5.64 ± 0.03	17.78 ± 0.06	9.25 ± 0.06	10.31 ± 0.09	4.45 ± 0.04
**Total flavonoids** **(TF)**	22.78 ± 0.77	19.81 ± 1.37	5.77 ± 0.38	10.12 ± 0.77	nd	nd	nd	0.50 ± 0.004	1.52 ± 0.02	7.80 ± 0.26	2.28 ± 0.14	2.43 ± 0.06
**Total hydrolysable tannins** **(THT)**	39.44 ± 1.91	26.23 ± 1.19	4.90 ± 0.29	9.06 ± 0.51	1.66 ± 0.06	1.00 ± 0.009	3.33 ± 0.08	1.89 ± 0.04	2.55 ± 0.04	19.82 ± 0.77	4.84 ± 0.11	2.68 ± 0.1
**Total condensed tannins** **(TCT)**	nd	nd	nd	nd	1.02 ± 0.01	1.17 ± 0.019	3.58 ± 0.14	1.67 ± 0.01	nd	nd	nd	nd
**Total phenolic compounds** **(TPC)**	70.92 ± 2.72 ^a^	53.97 ± 2.41 ^b^	20.49 ± 0.92 ^ef^	30.59 ± 1.2 ^d^	18.24 ± 0.03 ^f^	6.28 ± 0.09 ^g^	23.03 ± 0.26 ^e^	9.69 ± 0.03 ^g^	21.85 ± 0.01 ^ef^	36.87 ± 1.09 ^c^	17.42 ± 0.16 ^f^	9.56 ± 0.13 ^g^

^1^ Phenolic compounds tentatively identified according to the data shown in Table 1. ^2^ Quantitative data expressed as mg.g^−1^ of the freeze-dried extract are presented as mean ± standard deviation. nd: not detected. MAC-HE: maceration carried out with hydroethanolic solvent (80% EtOH); UAE-HE: Ultrasound-Assisted Extraction carried out with hydroethanolic solvent (80% EtOH); UAE-W: Ultrasound-Assisted Extraction carried out with water; MAE-W: Microwave-Assisted Extraction carried out with water. Different superscript letters in the same row indicate significant differences (*p* < 0.05, Tukey’s test).

**Table 3 foods-12-02596-t003:** Antioxidant capacity and cytotoxicity of *C. sativa* extracts.

Antioxidant Activity (EC_50_, mg.mL^−1^) ^1^	Extraction Method	Chestnut By-Products			Positive Control Trolox
		Leaves	Shells	Burs	
**TBARS**	MAC-HE	0.23 ± 0.05 ^b,c^	0.56 ± 0.03 ^d^	0.002 ± 0.0001 ^a^	0.0058 ± 0.0006
	UAE-HE	2.0 ± 0.2 ^e^	0.2 ± 0.01 ^b,c^	0.002 ± 0.0001 ^a^	
	UAE-W	0.36 ± 0.16 ^c^	0.018 ± 0.0005 ^a^	0.004 ± 0.0001 ^a^	
	MAE-W	0.08 ± 0.02 ^a,b^	0.071 ± 0.012 ^a,b^	0.008 ± 0.0002 ^a^	
**DPPH**	MAC-HE	0.22 ± 0.07 ^b^	0.12 ± 0.02 ^c^	0.33 ± 0.01 ^f^	0.043 ± 0.002
	UAE-HE	0.27 ± 0.02 ^b^	0.16 ± 0.01 ^a^	0.14 ± 0.02 ^c,d^	
	UAE-W	0.20 ± 0.12 ^a,b^	0.90 ± 0.08 i	0.28 ± 0.02 ^e,f^	
	MAE-W	0.18 ± 0.01 ^a,b^	0.07 ± 0.01 ^h^	0.32 ± 0.01 ^f,g^	
**Reducing Power**	MAC-HE	0.43 ± 0.06 ^e^	0.07 ± 0.01 ^d^	0.21 ± 0.02 ^a^	0.029 ± 0.003
	UAE-HE	0.14 ± 0.03 ^b^	0.17 ± 0.05 ^a,b^	0.28 ± 0.02 ^c^	
	UAE-W	0.17 ± 0.04 ^a,b^	1.13 ± 0.03 ^f^	1.34 ± 0.05 ^g^	
	MAE-W	0.18 ± 0.03 ^a,b^	0.20 ± 0.15 ^a^	0.32 ± 0.01 ^c^	
**Hepatotoxicity** (GI_50,_ μg.mL^−1^) ^2^					Elipticine
**PLP2 cells**	MAC-HE	>400 ^a^	>400 ^a^	>400 ^a^	1.4 ± 0.1
	UAE-HE	>400 ^a^	>400 ^a^	>400 ^a^	
	UAE-W	>400 ^a^	>400 ^a^	>400 ^a^	
	MAE-W	209 ± 19 ^c^	228 ± 10 ^b^	227 ± 5 ^a^	

Data are mean ± standard deviation (n = 9). ^1^ EC_50_ value refers to the extract or standard concentration (mg.mL^−1^) corresponding to 50% of antioxidant activity. ^2^ GI_50_ value refers to the extract or standard concentration (µg.mL^−1^) responsible for the inhibition of 50% of the growth of a primary culture of liver cells-PLP2. MAC-HE: maceration carried out with hydroethanolic solvent (80% EtOH); UAE-HE: Ultrasound-Assisted Extraction carried out with hydroethanolic solvent (80% EtOH); UAE-W: Ultrasound-Assisted Extraction carried out with water; MAE-W: Microwave-Assisted Extraction carried out with water. Different letters in the same assay indicate significant differences (*p* < 0.05, Tukey’s test).

**Table 4 foods-12-02596-t004:** Antibacterial (MIC and MBC) and antifungal (MIC and MFC) activity of extracts of *C. sativa* by-products.

	Leaves				Shells				Burs			
	MAC-HE	UAE-HE	UAE-W	MAE-W	MAC-HE	UAE-HE	UAE-W	MAE-W	MAC-HE	UAE-HE	UAE-W	MAE-W
	MIC/MBC	MIC/MBC	MIC/MBC	MIC/MBC	MIC/MBC	MIC/MBC	MIC/MBC	MIC/MBC	MIC/MBC	MIC/MBC	MIC/MBC	MIC/MBC
**Gram-negative bacteria**												
** *Enterobacter cloacae* **	2.5/10	5/>10	2.5/>10	1.25/>10	2.5/>10	2.5/>10	1.25/>10	5/>10	>10/>10	>10/>10	5/>10	2.5/5
** *Escherichia coli* **	10/>10	10/>10	5/>10	10/>10	10/>10	10/>10	5/>10	10/>10	>10/>10	>10/>10	5/>10	10/>10
** *Pseudomonas aeruginosa* **	10/>10	10/>10	10/>10	10/>10	>10/>10	>10/>10	5/>10	10/>10	>10/>10	>10/>10	>10/>10	2.5/5
** *Salmonella enterocolitica* **	5/10	5/>10	1.25/>10	1.25/>10	5/>10	5/>10	2.5/>10	5/>10	10/>10	10/>10	5/>10	1.25/10
** *Yersinia enterocolitica* **	1.25/10	1.25/10	5/>10	5/>10	1.25/>10	1.25/>10	10/>10	5/>10	>10/>10	>10/>10	>10/>10	>10/>10
**Gram-positive bacteria**												
** *Bacillus cereus* **	5/>10	5/>10	1.25/>10	2.5/>10	2.5/>10	2.5/>10	10/>10	5/>10	2.5/>10	2.5/>10	5/>10	1.25/>10
** *Listeria monocytogenes* **	2.5/10	5/>10	2.5/>10	1.25/>10	1.25/>10	1.25/>10	2.5/>10	5>10	10/>10	5/>10	1.25/>10	0.6/>10
** *Staphylococcus aureus* **	1.25/10	1.25/>10	2.5/>10	0.3/>10	0.6/>10	0.6>10	2.5/>10	5/>10	2.5/>10	1.25/>10	0.6/10	0.3/5
**Fungi**	**MIC** **/MFC**	**MIC** **/MFC**	**MIC** **/MFC**	**MIC** **/MFC**	**MIC** **/MFC**	**MIC** **/MFC**	**MIC** **/MFC**	**MIC** **/MFC**	**MIC** **/MFC**	**MIC** **/MFC**	**MIC** **/MFC**	**MIC** **/MFC**
** *Aspergillus brasiliensis* **	>10/>10	>10/>10	10/>10	>10/>10	>10/>10	>10/>10	>10/>10	>10/>10	10/>10	10/>10	10/>10	10/>10
** *Aspergillus fumigatus* **	>10/>10	>10/>10	10/>10	>10/>10	>10/>10	>10/>10	>10/>10	>10/>10	10/>10	10/>10	10/>10	10/>10

Results are expressed in mg.mL^−1^ extract and are presented as mean ± SD (*n* = 9). MIC: minimal inhibitory concentration; MBC: minimal bactericidal concentration; MFC: minimal fungicidal concentration; MAC-HE: maceration carried out with hydroethanolic solvent (80% EtOH); UAE-HE: Ultrasound-Assisted Extraction carried out with hydroethanolic solvent (80% EtOH); UAE-W: Ultrasound-Assisted Extraction carried out with water; MAE-W: Microwave-Assisted Extraction carried out with water. Positive controls (mg.mL^−1^): Streptomycin (1 mg.mL^−1^)—*E. cloacae* (0.007/0.007), *E. coli* (0.01/0.01), *P. aeruginosa* (0.06/0.06), *S. enterocolitica* (0.007/0.007), *Y. enterocolitica* (0.007/0.007), *B. cereus* (0.007/0.007), *L. monocytogenes* (0.007/0.007) and *S. aureus* (0.007/0.007 mg.mL^−1^); Methicillin (1 mg.mL^−1^)—*S. aureus* (0.007/0.007); Ampicillin (10 mg/mL)—*E. cloacae* (0.15/0.15), *E. coli* (0.15/0.15), *P. aeruginosa* (0.63/0.63), *S. enterocolitica* (0.15/0.15), *Y. enterocolitica* (0.15/0.15), *L. monocytogenes* (0.15/0.15), *S. aureus* (0.15/0.15); Ketoconazole (1 mg.mL^−1^)—*A. brasiliensis* (0.06/0.125); *A. fumigatus* (0.5/1).

## Data Availability

All related data and methods are presented in this paper. Additional inquiries should be addressed to the corresponding author.

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
