# Peer review of "Chemical and Bioactive Screening of Green Polyphenol-Rich Extracts from Chestnut By-Products: An Approach to Guide the Sustainable Production of High-Added Value Ingredients"

_foods, 2023, doi:10.3390/foods12132596_

Round 1
Reviewer 1 Report
The manuscript focuses on the valorization of chestnut by-products. The main aim the screening of the phenolic compound composition and bioactivities of different green extracts of chestnut burs, shells and leaves. The manuscript is well written reporting interesting results with sufficient novelty, significant contribution to knowledge and performance of this study.
Abstract: The authors should include the overall conclusion.
2.2. Plant material: The authors should provide particle size and moisture content of the sample material.
2.3.2. Ultrasound-assisted extraction (UAE): What about the temperature during UAE?
2.3.3. Microwave-assisted extraction (MAE): Which microwave power was used?
In Table 1, it is better to include the compound identification basis so as to highlight those identified for the first time, if any.
Is it really necessary the citation of the nine articles of the coauthor dr. Lillian Barros? Please, revise it.
Author Response
Reviewer #1: The manuscript focuses on the valorization of chestnut by-products. The main aim the screening of the phenolic compound composition and bioactivities of different green extracts of chestnut burs, shells and leaves. The manuscript is well written reporting interesting results with sufficient novelty, significant contribution to knowledge and performance of this study.
- Abstract: The authors should include the overall conclusion.
The suggestion was included. The conclusion was updated as follows: “Overall, it can be concluded that chestnut by-products, including burs, shells, and leaves, are sources of polyphenolic compounds with significant antioxidant and antimicrobial activities. The choice of extraction method and solvent greatly influenced the composition and bioactivity of the extracts. These findings highlight the potential of chestnut by-products for the development of natural additives, particularly for food preservation, while also emphasizing the importance of sustainable utilization of agricultural waste materials. Further research is warranted to optimize extraction techniques and explore additional applications for these valuable bioactive compounds.”
- Plant material: The authors should provide particle size and moisture content of the sample material.
The sample material was not sieved before the analysis, so we cannot say exactly what particle size was used. However, we always try to use a sample material as homogeneous as possible, and the particle size was approximately 0.5-1 mm. Regarding the moisture content present in the sample, unfortunately, this measure was not carried out, and we will take all these suggestions into account for future work. Ultrasound-assisted extraction (UAE): What about the temperature during UAE?
An ice bath was used to prevent the mixture from overheating, maintaining a temperature of approximately 20°C at the end of each extraction batch. This information has been added to the text.
- Microwave-assisted extraction (MAE): Which microwave power was used?
This information has been added to the text.
- In Table 1, it is better to include the compound identification basis so as to highlight those identified for the first time, if any.
The basis for compound identification is already in the text (lines 232-236). For compound identification of all peaks, we considered and interpreted all the combined information available, including the elution order on the C18 column and characteristics of the UV-Vis and mass spectra (molecular ion ([M-H]-), and MS/MS fragments), as well as the comparison with standards when available, literature data and libraries. Therefore, we have not modified the text, also because such information is also presented in detail in the result and discussion section, with the necessary citations.
- Is it really necessary the citation of the nine articles of the coauthor dr. Lillian Barros? Please, revise it.
Referencing all these articles of co-author Dr. Lillian Barros were indeed necessary as they are the reference to the methodologies developed by our research group and used in the analyses of this manuscript. All nine citations are related to methods, and then used also to contrast the results obtained using the same method. We understand that, if the case were self-citation, removal would be necessary.

Reviewer 2 Report
The manuscript refers to the chemical characterization of chestnut by-products in terms of polyphenol profile and content. Authors performed different extraction methods with different solvents. In adittion, authors included the study of in vitro antioxidant effects, hepatotoxicity and antimicrobial activity.
The aim of the manuscript is interesting and the methods described are properly utilized. However, manuscript could be improved:
-Line 27: "conventional solid-liquid extraction (MAC)". at this point, it is not clear why this extraction method is called MAC. Then in line 165 authors indicate that MAC refers to Maceration. Please include this word in the abstract.
-Line 32, 34, 41, 309, 311: Please use cientific notation correctly (superscripts). Also use subscripts for chemical compounds like in line 313. Control this in all the manuscript.
-Line 132: Indicate which phenolic compound standards.
-Line 227: Did the authors only have gallic acid, ellagic acid and quercetin 3-O-glucoside as references compounds?
-Figure 1: Please indicate figures with the letters A, B and C.
-Table 1: Include a column with the ployphenol fmaily compound clasification. In this footnote is not necessary to clarify the extraction methods abbreviation since they are not mentioned in the table.
-Polyphenol identification is really complete, however it is too long and confusing. I suggest to use sub-sections for each polyphenol family from line 371.
-Line 579: Letters that indicate significant differences are the same that authors use to clarify concepts. That is confusing.
-Figure 4: Include graphic legend
Figure 5: Include letters A and B to differentiate the figures.
Line 749: "absolute values g.g-1"? Or mg.g-1?
Table 3: Did the authors performed ANOVA test in this data? Did they found significat differences between extracts for each test?
Author Response
Reviewer #2: The manuscript refers to the chemical characterization of chestnut by-products in terms of polyphenol profile and content. Authors performed different extraction methods with different solvents. In addition, authors included the study of in vitro antioxidant effects, hepatotoxicity and antimicrobial activity. The aim of the manuscript is interesting and the methods described are properly utilized. However, manuscript could be improved:
- Line 27: "conventional solid-liquid extraction (MAC)". at this point, it is not clear why this extraction method is called MAC. Then in line 165 authors indicate that MAC refers to Maceration. Please include this word in the abstract.
Done.
- Line 32, 34, 41, 309, 311: Please use cientific notation correctly (superscripts). Also use subscripts for chemical compounds like in line 313. Control this in all the manuscript.
Done.
- Line 132: Indicate which phenolic compound standards.
Done
- Line 227: Did the authors only have gallic acid, ellagic acid and quercetin 3-O-glucoside as references compounds?
We have indeed forgotten to mention epicatechin. This information has been added to the text. The gallic acid and ellagic acid are fundamental components in the structure of hydrolysable tannins, which is why they are commonly used as reference standards for quantification. They can also be called ellagitannins and galotannins. The structure and physical-chemical characteristics of the other quantified compounds are similar to the chosen standard. In this case, quercetin-3-O-glucoside is a glycosylated flavonoid, as are isorhamnetin and kaempferol.
- Figure 1: Please indicate figures with the letters A, B and C.
Done.
- Table 1: Include a column with the ployphenol fmaily compound clasification. In this footnote is not necessary to clarify the extraction methods abbreviation since they are not mentioned in the table.
Done.
- Polyphenol identification is really complete, however it is too long and confusing. I suggest to use sub-sections for each polyphenol family from line 371.
Thanks for your suggestion. We place the class of phenolic compounds in Table 1. Thus, we believe that the table will be clearer and more explanatory.
- Line 579: Letters that indicate significant differences are the same that authors use to clarify concepts. That is confusing.
Thank you for your observation; the letters have been changed to numbers, so we believe that the concepts are clearer.
- Figure 4: Include graphic legend
Done.
- Figure 5: Include letters A and B to differentiate the figures.
Done.
- Line 749: "absolute values g.g-1"? Or mg.g-1?
Done. It is mg.g-1
- Table 3: Did the authors performed ANOVA test in this data? Did they found significat differences between extracts for each test?
Yes, we performed an ANOVA test on this data. We found significant differences between the extraction methods for each assay. The different letters assigned to each assay represent statistical significance at p<0.05.

Reviewer 3 Report
To whom it may concern,
The present manuscript entitled "Chemical and bioactive screening of green polyphenol-rich extracts from chestnut by-products: an approach to guide the sustainable production of high-added value ingredients" is a well-written manuscript, however, the connection of these results with the sustainable production needs to be discussed in the discussion section. Moreover, the conclusion section should be briefer. My specific feww corrections are shown in the pdf file.

Minor revision is suggested.
Author Response
Reviewer #3:
- The conclusion is excessively long.
The conclusion was diminished.
